# Secondary grain boundary dislocations alter segregation energy spectra

Xinren Chen [1], William Gonçalves [2], Yi Hu [1], Yipeng Gao[1], Patrick Harrison[3], Gerhard Dehm [1], Baptiste Gault [1,4], Wolfgang Ludwig[2], Edgar Rauch[3], Xuyang Zhou [1] ✉ & Dierk Raabe [1]

Grain boundaries (GBs) trigger structure-specific chemical segregation of solute atoms. According to the three-dimensional (3D) topology of grains, GBs - although defined as two-dimensional defects - cannot practically be free of curvature. This leads to discrete variations in the GB plane orientations. Topologically required arrays of secondary GB dislocations accommodate these variations as well as deviations from ideal coincidence site lattice GBs. We report here that these pattern-forming secondary GB dislocations can have an additional and, in some cases, even a much stronger effect on GB segregation than defect-free GBs. Using nanoscale correlative tomography combining crystallography and chemical analysis, we quantified the relationship between secondary GB dislocations and their segregation energy spectra for a model Fe-W alloy. This discovery unlocks design opportunities for advanced materials, leveraging the additional degrees of freedom provided by topologically-necessary secondary GB dislocations to modulate segregation.

Metals and alloys usually consist of crystalline grains that fill three-dimensional (3D) space. These grains meet at junctions to form piecewise two-dimensional defects known as grain boundaries (GBs). To reconcile force equilibrium at nodes and junctions with pore-free, space-filling topology of the non-platonic shaped grains[1], GBs in polycrystalline materials are seldom perfectly flat and therefore exhibit curvature[2] (Fig. 1a, b). At the microscale, GBs accommodate curvature via planar segments of differing inclination, with some segments containing nanoscale steps and a periodic array of secondary GB dislocations, thereby preserving lattice continuity[3–6], as illustrated in Fig. 1c. Secondary GB dislocations are well known to play particularly important roles in GB migration[7,8], sliding[9], and rotation[10].

Secondary GB dislocations also alter GB segregation. Impurities or solutes tend to segregate to GBs to minimize their total free energy, as originally described by the Gibbs adsorption isotherm[11–14]. Due to their localized strain fields, the pattern-forming secondary GB dislocations can induce additional—and sometimes even much stronger—GB segregation compared to the segregation in defect-free GB structures. Reported atomic scale investigations clarify the role of GB defects in solute segregation. For instance, Hu et al. showed that GB dislocations

and steps along Pt twin boundaries distorted local elastic fields and altered Au segregation[15–17], while Liebscher et al. observed enrichment of C, Fe, and N at specific facet junctions of faceted, tilted Si GBs[18]. These observations suggest that GB defect mediated segregation is widespread, yet its magnitude across diverse GB defect structures remains to be rigorously quantified experimentally. In addition, interactions between secondary GB dislocations and solute atoms significantly alter the mechanical and physical properties of materials[19]. Systematically characterizing these complex solute-GB interactions with highest spatial resolution and chemical sensitivity is critical for understanding GB decoration and unlocking atomic-scale engineering of alloys[20].

There are four major challenges for the experimental characterization of solute-GB interactions. Firstly, GBs often deviate from flat planes and exhibit undulating 3D structures[21,22]. Real GB structures invariably break down into planar segments of differing inclination, accompanied by regular patterns formed by secondary GB dislocations to accommodate topological constraints[1,2]. These secondary GB dislocations significantly alter solute segregation behavior and create local solute enrichment patterns[4,18,23–27]. To capture the full structural

[1]Max-Planck-Institut for Sustainable Materials, Düsseldorf, Germany. [2]Université Lyon I, MATEIS, INSA Lyon, CNRS UMR 5510, Villeurbanne, France. [3]Université Grenoble Alpes, Grenoble, France. [4]Department of Materials, Imperial College London, London, UK. ✉e-mail: x.zhou@mpi-susmat.de

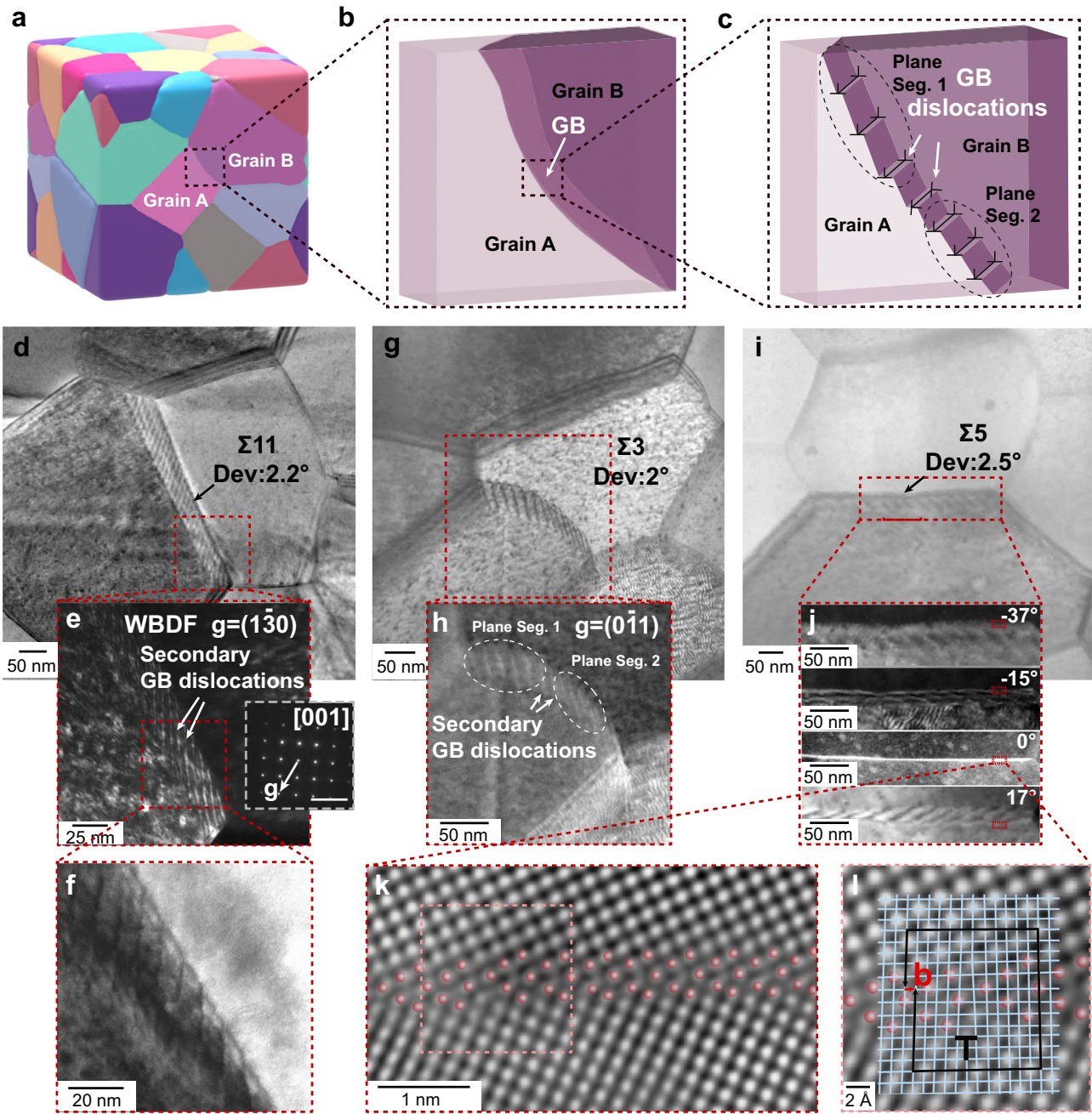

**Fig. 1 | Secondary GB dislocations in the Fe-1 at% W specimen. a** Illustration of grain boundaries (GBs) in a three-dimensional (3D) polycrystalline structure. **b** Magnification of a selected high-angle GB (HAGB) showing curved GB plane. **c** Illustration of steps and secondary GB dislocations on the GB. **d** Bright-field image of a transmission electron microscopy (TEM) lamella prepared from the body-centered cubic (BCC) Fe-1 at% W specimen, featuring multiple GBs, one of which is labeled as a HAGB. This HAGB is a Σ11 GB that deviates by 2.2° from the theoretical misorientation. **e** Corresponding weak-beam dark-field (WBDF) image of the same area, imaged under the two-beam condition with **g** = [1̄30]. The inset image in the right side of e displays the diffraction pattern collected from the left grain ([100] zone axis). Scale bar: 1/0.1 nm. **f** High magnification bright-field image displaying the steps and secondary GB dislocations of the GB. **g** Bright-field image of a Σ3 GB with a 2° deviation from the ideal misorientation. **h** Corresponding dark-field image of the Σ3 GB with **g** = [01̄1], revealing secondary GB dislocations. **i** Bright-field image of a Σ5 GB that deviates by 2.5° from the ideal misorientation. **j** Dark-field images of the Σ5 GB acquired at various α tilt angles. Scale bar: 50 nm. **k** Magnified view of the area at 0° tilt angle, with semi-transparent red dots highlighting the atomic column configuration near the GB. See additional details in Supplementary Fig. 5. **l** The region in k overlaid with a displacement shift complete (DSC) lattice and a Frank circuit, illustrating the presence of a secondary GB dislocation characterized by the Burgers vector (**b**).

and chemical characteristics of a GB, one must conduct these measurements in 3D. Accurate quantification of solute segregation requires atomic-scale spatial resolution and high chemical sensitivity, as solute segregation is confined, typically to only a few atomic layers at the GB[12–14,28–30]. In addition, a complete representation for any GB requires the measurement of five kinematic degrees of freedom[31]:

three to describe the misorientation between adjacent grains and two to define the orientation of the GB segment plane. Finally, different types of GBs exhibit different segregation behaviors, sometimes varying by more than an order of magnitude. To conduct a statistically relevant analysis, it is essential to have the capability to study multiple GBs with a variety of misorientations and GB plane orientations, ideally

simultaneously. This approach allows for statistical relevance and is one order of magnitude faster than the conventional method of analyzing a single GB per data set.

In this work, we apply four-dimensional scanning transmission electron microscopy (4DSTEM) tomography[32] to gain structural insights and use atom probe tomography (APT) for compositional analysis on a model body-centered cubic (BCC) Fe-1 at% W alloy. This correlative nanoscale tomography approach enables the measurement of five kinematic degrees of freedom for any GB[31] and quantifies near atomic-scale solute-GB interactions at defect-rich GBs. 4DSTEM tomography efficiently detects defect contrasts at GBs, eliminating the need for meticulous crystal tilting required by conventional transmission electron microscopy (TEM) and simplifying the characterization of randomly oriented GBs with undulating 3D structures. By examining both the structural and chemical characteristics across 12 Fe GBs, we provide quantitative analyses for revealing the correlations between GB structure and W segregation. This knowledge is essential for quantifying the interactions between secondary GB dislocations and alloying elements, opening design opportunities for atomic engineering of future materials. For instance, tailoring texture to modulate secondary GB dislocations can be used to further adjust solute segregation through localized elastic interactions and ultimately enhance ductility, impact strength, and fracture toughness.

## Results and discussion

### Correlative tomography for three-dimensional crystallography and chemistry

We prepared the model BCC Fe-1 at% W alloy thin film using physical vapor deposition (PVD) accompanied by a heat treatment at 500 °C for 240 min to activate diffusion and facilitate solute segregation at GBs. The average grain size is approximately 134 nm ± 50 nm, with equiaxed grains and a weak {111} fiber texture in the growth direction (see Supplementary Fig. 1). Periodic intensity contrasts in the weak-beam dark-field (WBDF) TEM images of high-angle GBs (HAGBs) typically stem from secondary GB dislocations, particularly in those nearly matching the coincidence site lattice (CSL) relationship[33]. Figure 1d–f present the bright-field and WBDF images of a $\Sigma 11$ GB that has a 2.2° deviation from the ideal misorientation. Under two-beam conditions with $\mathbf{g} = [1\bar{3}0]$, periodic contrast appears along the GB. We attribute this contrast to secondary GB dislocations (see Fig. 1e). Additional information from high-angle annular dark-field (HAADF) STEM imaging (see Supplementary Fig. 2) further reveals the secondary GB dislocation network and the associated local lattice distortions.

To provide an independent check, we acquired 4DSTEM data and generated virtual dark-field images (see Supplementary Figs. 3 and 4), which reproduce the contrast observed in conventional WBDF images. This demonstrates that secondary GB dislocations are consistently detected by both techniques and confirms their role in producing the observed intensity modulation.

Figure 1g, h show a similar analysis for a $\Sigma 3$ GB with a 2° deviation, where secondary GB dislocations are also observed under two-beam conditions with $\mathbf{g} = [0\bar{1}1]$. The GB can be decomposed into a series of planar segments, each characterized by a distinct density of secondary GB dislocations. Figure 1i presents the periodic contrast features arising from secondary GB dislocations in a $\Sigma 5$ GB exhibiting a 2.5° deviation, while Fig. 1j shows these features under varying tilt angles. Figure 1k provides atomic-resolution images of the selected region in Fig. 1j, capturing the termination point of a secondary GB dislocation in the $\Sigma 5$ GB. The atomic columns around the GB are highlighted with red markers (also see Supplementary Fig. 5). We overlay the observed structure with a displacement shift complete (DSC) lattice and construct a Frank circuit, as shown in Fig. 1l (see Supplementary Fig. 6 for the reference circuit), to determine the projected Burgers vector along the beam direction associated with the secondary GB dislocation.

To further elucidate the interactions between solutes and defects (specifically secondary GB dislocations), we employed our recently introduced nanoscale tomography approach[31,32]. Figure 2a-c show a 3D crystallographic reconstruction of 11 grains and 12 GBs in a needle-shaped Fe-W specimen, providing a meaningful reference for the GB segregation landscape in materials with defect-containing GBs. We colored each grain based on its Euler angle representation of the crystallographic orientation relative to the Z-axis (see Supplementary Table 1), along which the specimen is tilted for tomography (see Fig. 2a), with the X-axis aligned parallel to the thin film growth direction. The orientation was determined by analysis of the nanobeam diffraction patterns that were systematically collected during the 4DSTEM scans. Figure 2b displays three example diffraction results taken at different tilts. Our method enables the mapping of local normal to the GB plane between adjacent grains, as demonstrated in Supplementary Fig. 7. Details on the sample preparation and data processing are available in the Sample preparation, 4DSTEM tomography, Supplementary Fig. 8a, and Supplementary Movie 1.

We analyzed the composition of this same specimen by APT. Figure 2d displays the spatial distribution of Fe as individual red dots, and a set of isocomposition surfaces that delineates regions containing 5 or more at% W. The W enrichment closely correlates with the location of GBs identified in the 3D crystallographic reconstruction in Fig. 2c, evidencing preferential W segregation to GBs. We focus particularly on the following two GBs with a near-$\Sigma$ relationship that exhibit discontinuous W segregation patterns.

### Quantitative analysis of secondary grain boundary dislocations and chemical segregation

To quantify segregation patterns, a 3D compositional mapping of W within the APT reconstruction was calculated with a resolution of $0.5 \times 0.5 \times 0.5\,\text{nm}^3$ and a delocalization parameter of $3\,\text{nm}$[34]. The compositional map plotted in Fig. 3a shows that the segregation of W at GBs forms a discontinuous pattern rather than a uniform planar distribution. We selected two GBs for detailed analysis: GB1 between grain $\alpha_1$ and grain $\alpha_2$, characterized as a near $\Sigma 5$[100] GB with a tilt deviation of 1.9° along [111] from its ideal misorientation, and GB2 between grain $\alpha_4$ and grain $\alpha_5$, identified as a near $\Sigma 13b$[111] GB with a 0.6° tilt deviation along [111], illustrated in Fig. 3b–e, f–i, respectively.

Figure 3 b presents two mappings of the local normal to the GB plane between adjacent grains for GB1: the bottom mapping corresponds to grain $\alpha_1$, and the top to grain $\alpha_2$. The character of GB1 corresponds to a geometrically curved boundary comprising a range of local orientations rather than an ideal $\Sigma 5$ symmetric tilt or twist GB. The color variation along GB1 reflects this orientation spread.

We identified secondary GB dislocations by constructing a virtual dark-field image based on the vector $\mathbf{g} = [12\bar{1}]$ for grain $\alpha_2$, as shown in Fig. 3c. See more details in Supplementary Fig. 9. Figure 3d displays a stairwell-like periodic segregation pattern at this GB, highlighted by a set of isosurfaces with a threshold of 2.5 at% W. A one-dimensional (1D) composition profile with a bin size of 1 nm was calculated along the cylinder positioned through the segregation pattern in Fig. 3d. From this profile, plotted in Fig. 3e, we quantify the level of segregation ranging from 1 to 4 at% W. Note that this pattern exhibits a regular spacing of approximately 22 nm, a spacing that closely matches the spatial distribution of secondary GB dislocations imaged by virtual dark-field in Fig. 3c. The maximum value in the local composition profile is twice that of the mean composition value of the GB. This direct correlation suggests a key role of secondary GB dislocations in altering solute segregation, in terms of mechanism, trapping depth, patterning, magnitude and kinetics.

We observe similar results at the $\Sigma 13b$[111] GB, as shown in Fig. 3f–i. Unlike the previous case, GB2 is an asymmetric tilt GB, with the plane normal for grain $\alpha_4$ aligning with $(\bar{1}\bar{1}2)$ and for grain $\alpha_5$ with

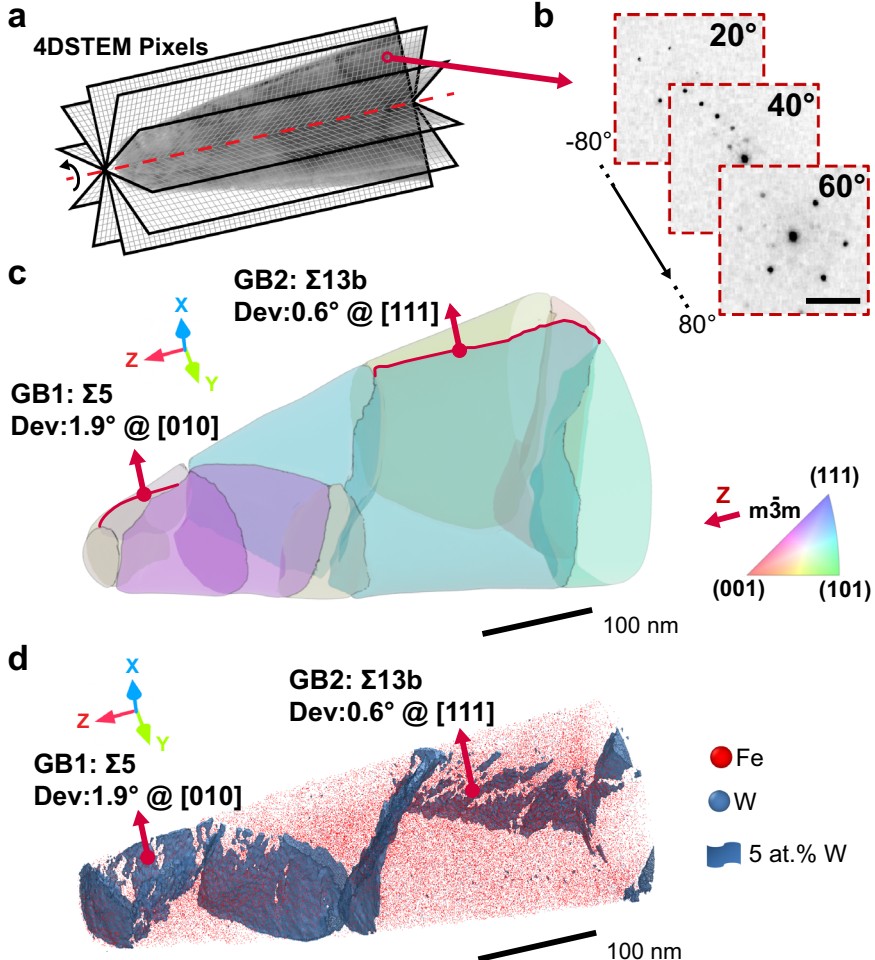

**Fig. 2 | Correlative tomography characterization of the Fe-1 at% W specimen.**
**a** Illustration of four-dimensional scanning transmission electron microscopy (4DSTEM) tomography. Each pixel in the 4DSTEM data incorporates a local nano-beam diffraction pattern. **b** Representative nanobeam diffraction patterns from the 4DSTEM datasets at different tilt angles for the grain indicated by red circle in a. Scale bar: 1/0.083 nm. **c** 3D crystallographic reconstruction of the grains in the Fe-W needle-shaped specimen, characterized via 4DSTEM tomography[31,32]. Grains are colored based on the Euler angle representation of their crystallographic orientations relative to the Z-axis, serving as the tilting axis in TEM. The X-axis represents the thin film growth direction in the same coordinate system in Supplementary Fig. 1. **d** 3D chemical reconstruction of the same specimen shown in c by atom probe tomography (APT), illustrating the spatial distribution of Fe (red) and W (cornflower blue) atoms with a superimposed 5 at% W isosurface. The 5 at% isosurface represents the region (voxels) containing 5 or more at% W. For a clear visualization, we aligned the APT reconstruction in the same perspective as the 4DSTEM tomography reconstruction shown in c.

($\bar{1}$10), as detailed in Fig. 3f. Nevertheless, we still find a close match between the secondary GB dislocations shown in Fig. 3g and Supplementary Fig. 10, and the periodic segregation patterns (see Fig. 3h) and the composition profile plotted in Fig. 3i. We also observed GB facets that modulate solute segregation at GBs, consistent with previous reports[18,24,27,35]. These were particularly noted at GB3 (see Supplementary Fig. 11 for detailed analysis).

**Calculation of grain boundary segregation energy using the Langmuir-McLean isotherm**
The relationship between solute segregation and segregation energy typically follows the Langmuir-McLean isotherm, as detailed in Equation (1)[11,36–38],

$$\frac{X_{\text{GB}}}{1 - X_{\text{GB}}} = \frac{X_{\text{B}}}{1 - X_{\text{B}}} \exp\left(\frac{\Delta G_{\text{Seg}}}{RT}\right). \quad (1)$$

Here, $\Delta G_{\text{Seg}}$ is the segregation energy of the solute atom, $R$ is the ideal gas constant, $T$ is the temperature, and $X_{\text{B}}$ and $X_{\text{GB}}$ are the compositions of solute in the bulk and at the GB, respectively. $X_{\text{GB}}$ can be further expressed as $\frac{\Gamma \cdot \Omega (1 - X_{\text{B}})}{t} + X_{\text{B}}$[39], where $\Gamma$ represents the interfacial

excess (IE), denoting the excess number of atoms per unit area at an interface, $t$ is the thickness of the GB, and $\Omega$ is the atomic volume of the solvent.

We mapped the IE values across all visible GBs in the APT dataset, with Fig. 4a providing examples of the integral profiles across GB1 used for these IE calculations, and Fig. 4b showing the mapped IE values[40,41]. We chose IE as it minimizes the influence of local magnification caused by different field evaporation behaviors arising from different compositions and grain orientations[41,42]. Solute segregation at GBs typically spans only a few atomic layers, which is often smaller than the volume increment, potentially leading to inaccurate measurements of local segregation. Consequently, IE is employed for accurate quantification analysis.

Figure 4c maps the segregation energy calculated from the IE, based on the Langmuir-McLean isotherm, Eq. (1). While this isotherm does not account for solute-solute interactions, for more complex cases, models such as the Fowler-Guggenheim isotherm[43] or the Seah-Hondros isotherm[44] could be similarly applied by adding an extra term to account for the interaction energy. Despite these complexities, we hypothesize that the general formulation suggested by the Langmuir-McLean isotherm (Eq. (1)) remains applicable here, except for the case

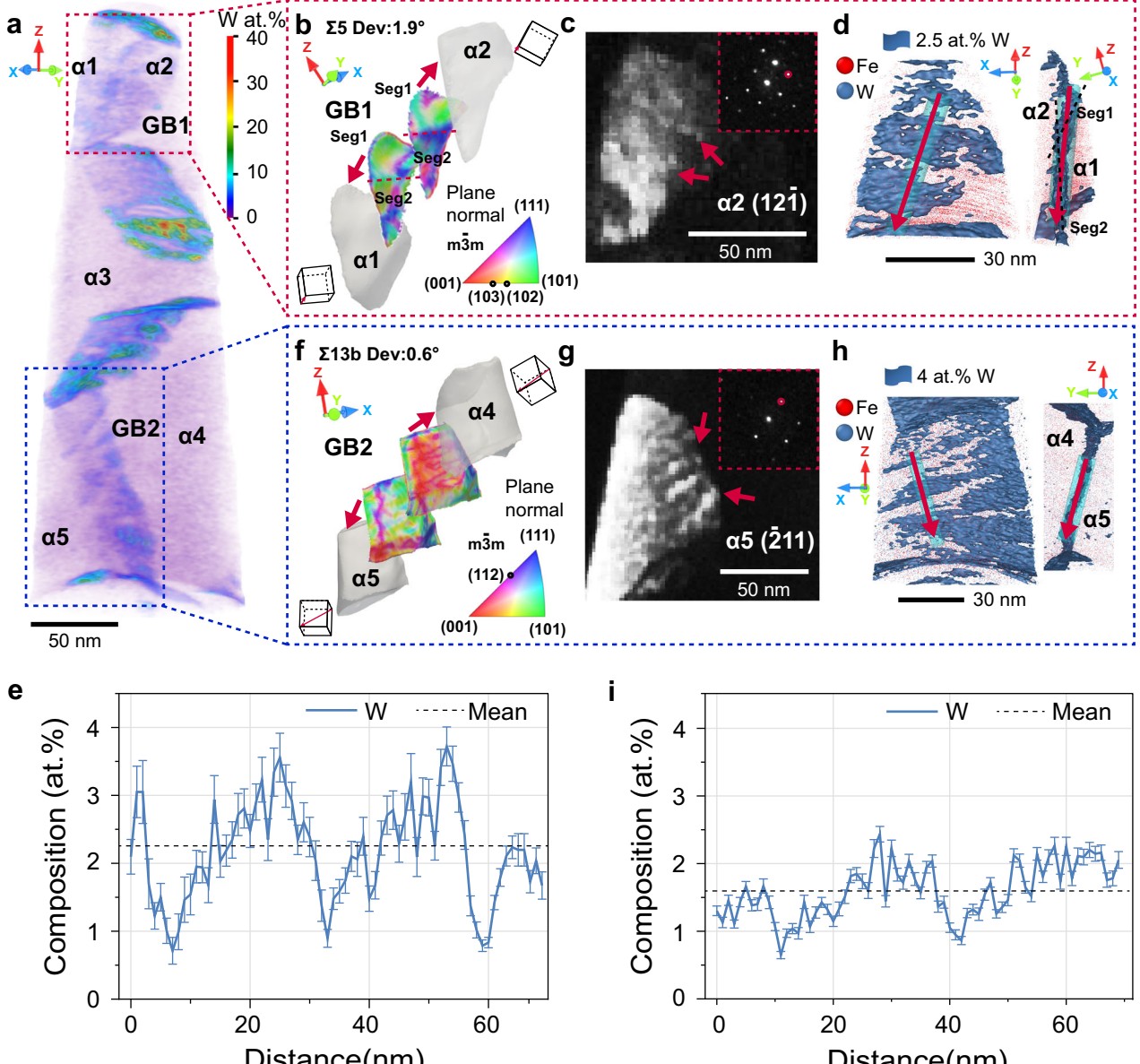

**Fig. 3 | Characterization of secondary GB dislocations and their linkage to segregation patterns in the Fe-1 at% W specimen. a** Compositional mapping of W at a resolution of $0.5 \times 0.5 \times 0.5$ nm³ per voxel, enabling visualization of volume-specific variations. Parts of the grains and GBs have been labeled, with identities ranging from grain $\alpha_1$ to grain $\alpha_5$ and from GB1 to GB2 (see all labels in Supplementary Fig. 7). The X-axis represents the thin film growth direction in the same coordinate system in Supplementary Fig. 1. Correlative crystallographic and compositional quantitative analysis for two GBs GB1: **b**–**e** and GB2: **f**–**i**. Each set includes: **b**, **f** orientation mappings of the local normal to the GB plane between adjacent grains (rendered in translucent gray), with two mappings provided for each GB, referenced to the respective grain involved. Cubic symbols in indicate the orientation of the grains and the red arrow indicates the misorientation rotation axis. **c**, **g** 4DSTEM virtual dark-field images; **d**, **h** atom maps of Fe (colored red) and W (colored cornflower blue), with superimposed isosurfaces at 2.5 at% W and 4.0 at% W, respectively; **e**, **i** W profiles along the red arrows shown in (**e**, **i**). The insert images in (**c**, **g**) display the nanobeam diffraction patterns with the vectors **g** = [12$\bar{1}$] for grain $\alpha_2$ and **g** = [$\bar{2}$11] for grain $\alpha_5$ (highlighted by red circles), which were used to generate the virtual dark-field images shown in (**c**, **g**). Two perspectives (90° rotated clockwise from left to right) in (**d**, **h**) are shown to illustrate the segregation patterns in 3D.

that it cannot account for any second order defect trapping on the GB, such as caused by GB dislocation structures.

We calculated the mean segregation energy for each GB. As plotted in Fig. 4d, it shows the same trends previously reported from low angles to 60°[45], demonstrating a clear effect of misorientation on the segregation level. Our measurements allow for much more refined analysis of the segregation energy landscape. We have generated segregation energy spectra, Fig. 4e–g for GB1, GB2, and all 12 GBs. The statistical distribution of solute segregation is generally close to a skew-normal function (see Fig. 4e, f for GB1 and B2), as predicted by atomistic simulations[37], a feature that had not yet been confirmed

before by experiment. The experimental segregation energy spectra of defect-containing GBs deviate from the ideal curve, as marked by black arrows in Fig. 4e, f. It is noteworthy that this minor deviation can lead to peak splitting into two distinct domains in the segregation energy spectrum, as shown in Supplementary Fig. 12, when readily visible facets appear in GBs (see the facets in Supplementary Fig. 11).

## Estimation of the elastic energy contribution for grain boundaries containing secondary grain boundary dislocations

To identify the impact of secondary GB dislocations on solute segregation, we analyzed the periodic variations in segregation energy

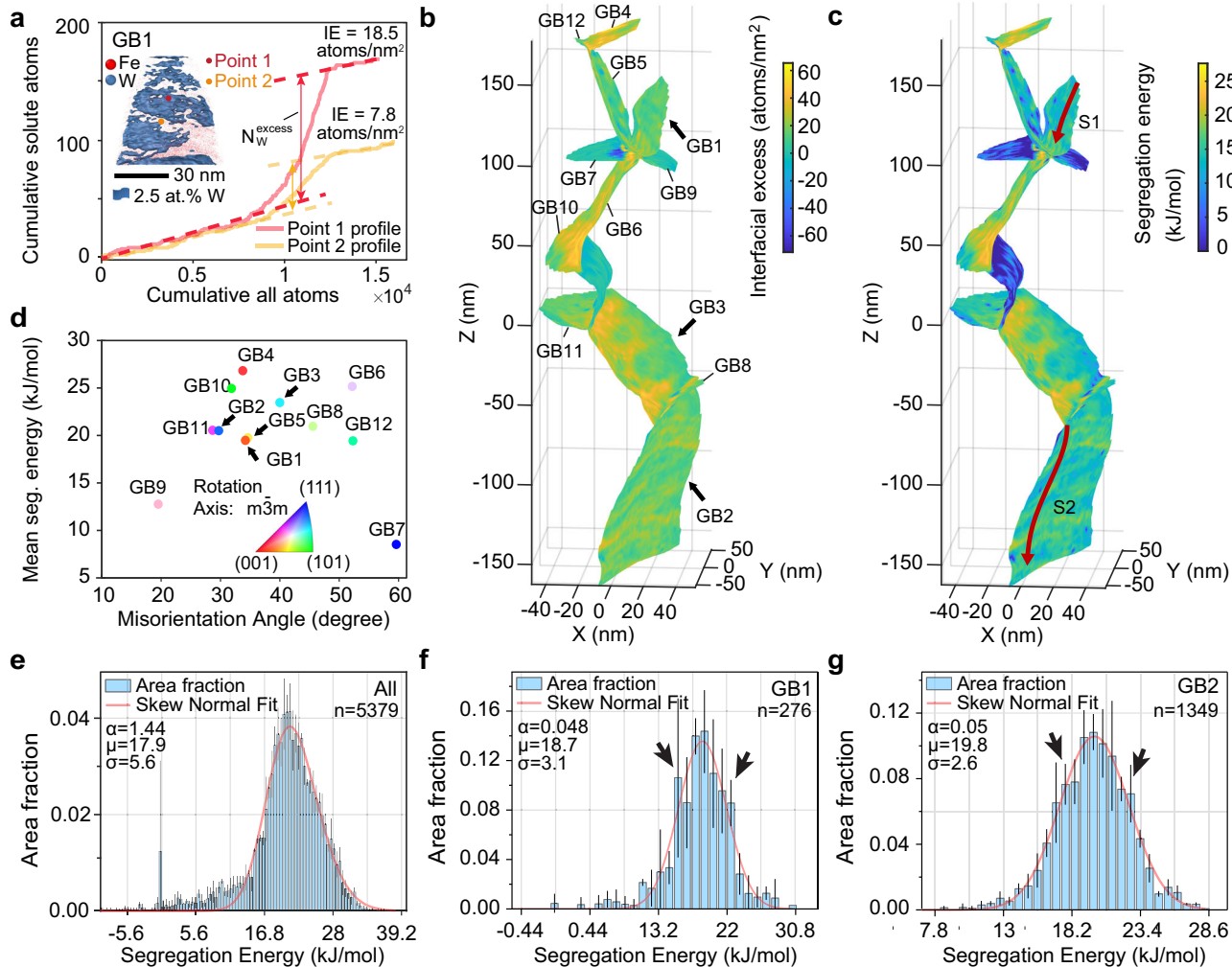

**Fig. 4 | Mapping and analysis of GB properties in the Fe-1 at% W specimen.**
**a** Integral profiles across GB1 for quantifying the interfacial excess (IE) of W segregation at the Fe GB[40,41]. We plot measurements from two points: Point 1 for high W regions (colored pink) and Point 2 for low W segregation regions (colored orange). In each plot, the solid lines show the cumulative relationship between all atoms and solute atoms, while the dashed lines are the fittings within the two grains adjacent to the GB plane. $N_W^{excess}$ represents the accumulation of excess atoms across the GB interface. IE values are calculated by dividing $N_W^{excess}$ by the corresponding interface areas, which are approximately 8.0 nm² for Point 1 and Point 2. We indicate the locations of Point 1 and Point 2 in the embedded image, consistent with Fig. 3d. **b** IE mapping on GBs, as identified from the correlative characterization in Fig. 2c, d. We labeled the locations of all investigated GBs from GB1 to GB12 in (**b**). **c** Segregation energy mapping on the same GBs as in b, calculated based on the Langmuir-McLean isotherm, Eq. (1)[11,36]. The locations where 1D line profiles of segregation energy values were measured are marked by red lines S1 and S2, details

of which will be presented in the following section. **d** Correlation between the misorientation and segregation energy of all investigated GBs: This scatter plot displays the average segregation energy (in kJ/mol) for various GBs, plotted against their misorientation (in degrees). Each point represents a different GB (GB1–GB12), with colors indicating the rotation axis of misorientation related to the cubic crystal symmetry ($m\bar{3}m$), as shown in the color triangle legend. Experimental GB segregation energy spectra for **e** GB1, **f** GB2, and **g** all GBs. Here, *n* represents the sample size used to derive statistics. The fitting plots, overlaid onto the histograms, generally follow the skew-normal function[37] but with local deviations highlighted by the black arrows, see (**e, f**). The used fitting parameters are: characteristic energy ($\mu$) in kJ/mol, width ($\sigma$) in kJ/mol, and shape parameter ($\alpha$) along with each plot. The sharp peak appearing near 0 kJ/mol in (**g**) results from the contribution of the GBs without solute segregation and the inevitable minor inclusion of the bulk regions adjacent to the GBs.

along GB1 and GB2 (see Fig. 4c, Lines S1 and S2), using 1D composition profiles plotted in Fig. 5a, b. The gradient in these profiles can be attributed to the secondary GB dislocations visible in Fig. 3c, g. The magnitude of the segregation energy change caused by these secondary GB dislocations can reach up to approximately 6 kJ/mol (corresponds to 62 meV/atom). It is important to note the positions of the troughs and peaks in Fig. 5a, b. For GB1, these are located around 16 kJ/mol and 20 kJ/mol, and for GB2 at 14 kJ/mol and 23 kJ/mol. These positions correspond precisely to the prominent shoulders marked by black arrows in Fig. 4f, g, providing side evidence for explaining the deviation of the segregation energy spectra from the skew-normal distribution.

Secondary GB dislocations generate a stress field that is nonlinear and contains non-elastic contributions within its core region, and is linear and elastic outside of the core. The dislocation core, which normally spans the length of two Burgers vectors[46], serves as an essential trap for solutes. Notably, the core size is smaller than the binning scale used for our segregation energy quantification, which is approximately 2 nm (see the line profile in Fig. 5a, b). Our main goal in this paragraph is to understand the gradient in these composition profiles. Thus, our analysis only focuses on the linear and elastic part of the stress field associated with the secondary GB dislocations. The significant impact of secondary GB dislocations on solute segregation necessitates the introduction of an additional term to accurately

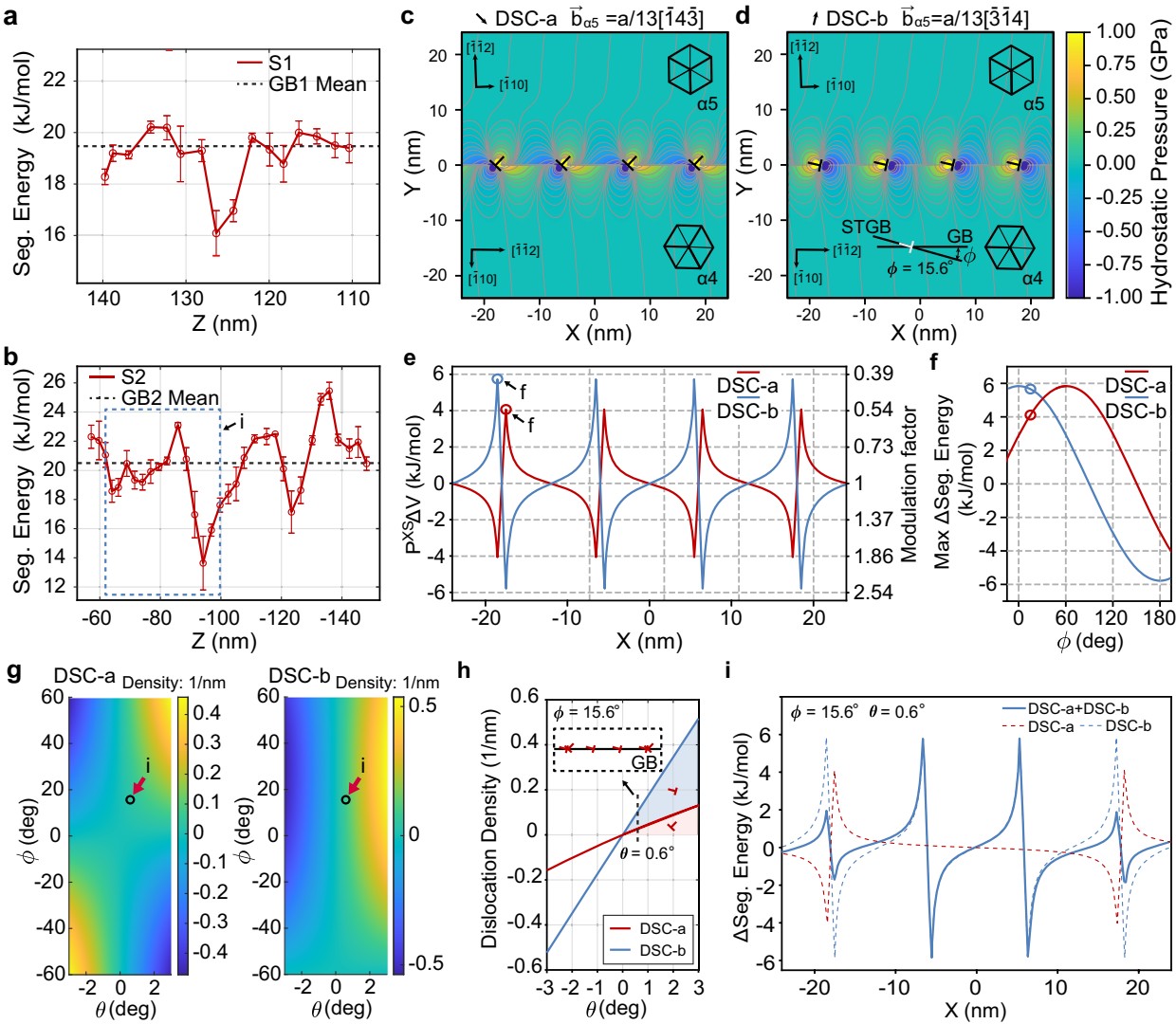

**Fig. 5 | Quantifying elastic energy for the Σ13b GB with dislocations using linear anisotropic elasticity theory. a**, **b** Segregation energy profiles along the lines marked by red arrows in Fig. 4c: **a** Line S1 at GB1 and **b** Line S2 at GB2, illustrating variations in segregation energy along these GBs. Horizontal dashed lines in a and b indicate the mean segregation energy of GB1 and GB2, respectively, serving as references. **c**, **d** The stress field surrounds secondary GB dislocations for GB2 (Σ13b, see Fig. 3f–i) with the DSC-lattice vectors as Burgers vectors: **c** DSC-a: $\mathbf{b}_{\alpha5} = \frac{a}{13}[\bar{1}4\bar{3}]$ and **d** DSC-b: $\mathbf{b}_{\alpha5} = \frac{a}{13}[\bar{3}\bar{1}4]$. Here, $a$ is the lattice constant of BCC Fe. **e** The change in segregation energy ($P^{XS}\Delta V$) aligns along the GB (averaged within a 0.5 nm distance from the GB plane), corresponding to (**c**, **d**). Here, DSC-a and DSC-b have the same Burgers vectors as those stress fields in (**c**, **d**). The modulation factor is defined as $\exp\left(\frac{-P^{XS}\Delta V}{RT}\right)$. **f** Maximum change in segregation energy (ΔSeg. Energy) caused by DSC-a or DSC-b dislocations as a function of inclination $\phi$. The blue and red circles indicate the experimentally relevant inclinations. **g** Dislocation density maps for DSC-a and DSC-b types as a function of inclination $\phi$ and misorientation deviation $\theta$ from the ideal Σ13b GB (rotation axis [111]), derived using the Frank-Bilby equation. **h** Dislocation densities for DSC-a and DSC-b as a function of misorientation deviation $\theta$ at a fixed inclination of $\phi = 15.6°$. **i** Calculated segregation energy profiles for the Σ13b GB with a misorientation deviation of $\theta = 0.6°$, showing contributions from DSC-a, DSC-b secondary GB dislocations, and their combined effect.

represent the change in segregation energy. The total segregation energy at the GB is expressed as:

$$\Delta G_{seg} = \Delta G_{seg}^{in} - P^{XS}\Delta V - T\Delta S_{seg}^{XS}. \tag{2}$$

Here, $\Delta G_{seg}^{in}$ represents the intrinsic segregation energy of the solute at the GB, excluding contributions from secondary GB dislocations and accounting for the sum of the segregation enthalpy, elastic energy, and segregation entropy[47], $P^{XS}$ denotes the hydrostatic pressure caused by the presence of secondary GB dislocations, $\Delta V$ represents the change in volume required for solute atoms to replace solvent atoms, and $\Delta S_{seg}^{XS}$ refers to the change in excess segregation entropy due to the introduction of secondary GB dislocations. $\Delta G_{seg}^{in}$ depends on the kinematic degrees of freedom for a given GB[48,49], while

$\Delta S_{seg}^{XS}$ reflects the change in the configurations due to the presence of secondary GB dislocations. Computational work by Tuchinda et al. demonstrates that in most alloy systems, the entropy contribution typically alters the segregation energy by less than 15%[50]. We consequently neglect the $\Delta S_{seg}^{XS}$ term and primarily attribute the modulation of segregation energy within a GB to the elastic energy resulting from secondary GB dislocations.

For bulk dislocations, the surrounding stress field can induce solute segregation in the areas surrounding the dislocations, known as Cottrell atmospheres[51–54]. In the case of secondary GB dislocations, the same applies, leading to a similar solute segregation phenomenon. Unlike Burgers vectors in the bulk lattice, the unit Burgers vectors at GBs[4] are known as the DSC vectors, shown in Supplementary Table 2 for GB2. The pronounced contrast in secondary GB dislocations, seen

in Fig. 3c, g, suggests the formation of GB ledges through the agglomeration of individual secondary GB dislocations, as documented in previous studies[55,56]. The tensile and compressive hydrostatic stress fields adjacent to secondary GB dislocations or GB ledges are the primary reason for the highest and lowest points in the 1D composition profiles plotted in Fig. 3e, i and Fig. 5a, b. The pronounced compositional slope indicates that this undulation is likely caused by stress gradients associated with secondary GB dislocations, which will be theoretically estimated in the following paragraphs.

We chose GB2 between grain $\alpha_4$ and grain $\alpha_5$ ($\Sigma13b$) to calculate the elastic field around secondary GB dislocations (see Supplementary Figs. 13 and 14) using the Stroh formalism for anisotropic elasticity theory[57–59] (see "Stress field simulation"). The Stroh formalism requires a cutoff distance near the dislocation core, beyond which linear anisotropic elasticity theory applies and within which it fails. Here, a cutoff of 0.5 nm was used for the stress field simulations. It is worth mentioning that non-linear anisotropic elasticity calculations by Lazar et al.[46] and atomistic simulations by Clouet et al.[60] are theoretical methods for estimating the stress field in the dislocation core. Inputs for the stress field calculations were obtained directly from our 4DSTEM tomography analysis, i.e., the five kinematic degrees of freedom for GB2 (see Supplementary Figs. 7a, b and Supplementary Table 3). Figure 5c, d show a periodic pattern in the stress field around the secondary GB dislocations with the DSC-lattice vectors $\mathbf{b}_{\alpha5} = \frac{a}{13}[1\bar{4}3]$ and $\mathbf{b}_{\alpha5} = \frac{a}{13}[\bar{3}1\bar{4}]$ function as the Burgers vectors[61]. Here, $a$ is the lattice constant of BCC Fe.

The orientation of the Burgers vector significantly influences the elastic field around the GB cores by varying its intensity and spread (see Fig. 5c, d). With the formula, $P^{XS}\Delta V$, we further calculated the extra contribution to the segregation energy from the elastic field produced by secondary GB dislocations, as presented in Fig. 5e for the quantitative measurements along the GB plane. The elastic energy contributes a modulation factor ranging from 0.4 to 2.5 times relative to the prediction of the Langmuir-McLean isotherm, corresponding to attenuation or enhancement depending on whether it takes a value below or above unity, respectively.

Besides modulating local solute distribution, secondary GB dislocation formation creates high-energy segregation sites at the dislocation cores and introduces additional solute into GBs[18,45]. Herbig et al. demonstrated this through correlative TEM-APT analysis, showing that deviations from ideal CSL misorientations increase mean solute segregation[45]. Our experimentally measured mean solute segregation includes the additional contribution from high-energy segregation sites associated with secondary GB dislocations. This work advances previous studies by quantifying and interpreting the heterogeneous chemical distribution at defective GBs. In the following, we further strengthen the theoretical understanding by investigating how GB crystallography influences secondary GB dislocation structures and the corresponding in-plane distribution of solute W.

Figure 5f shows the maximum change in segregation energy induced by DSC-a or DSC-b dislocations as a function of the inclination $\phi$, where $\phi$ defines the local deviation of the GB plane from its symmetric orientation. For instance, at $\phi = 0°$, the DSC-b dislocation generates a peak segregation energy increase of approximately +6 kJ/mol. As $\phi$ increases to $\phi = 90°$, the elastic contribution of the DSC-b dislocation to the segregation energy progressively decreases.

To evaluate how GB crystallography influences segregation, we calculated the densities of DSC-a and DSC-b dislocations as functions of inclination $\phi$ and misorientation deviation $\theta$ from the ideal $\Sigma13b$ GB (rotation axis [111]), using the Frank-Bilby equation[62,63] (see "Secondary grain boundary dislocation density calculation"), as shown in Fig. 5g. Figure 5h demonstrates that both dislocation densities increase with misorientation deviation $\theta$, with DSC-a and DSC-b exhibiting distinct trends at a fixed inclination of $\phi = 15.6°$, characteristic of the $\Sigma13b$ GB shown in Fig. 5h inset. Assuming that DSC-a and DSC-b dislocations are

each arranged with uniform spacing and positioned independently along the GB, based on the calculated dislocation density, we reconstructed the segregation energy landscape shown in Fig. 5i. The superposition of their elastic fields results in partial cancellation and produces a periodic modulation along the GB. Notably, small secondary spikes appear between the main peaks and troughs, consistent with the fine structure observed experimentally in the blue dashed box in Fig. 5b.

Our experiments confirm that the formation of secondary GB dislocations impacts W segregation at GBs in BCC Fe, resulting in an up to 100% increase in the composition of GB segregation compared to a defect-free appearing GB segment. Such secondary GB dislocations form patterns with regular spacing, which are topologically necessary to accommodate both the deviation from the ideal CSL misorientation and local variations in the GB plane orientation.

We have successfully quantified the segregation energy spectra of these GBs using IE mapping from APT data. The deviation in segregation energy within these spectra can reach magnitudes of up to approximately 6 kJ/mol. The extra elastic energy caused by secondary GB dislocations introduces a modulation factor, for example, ranging from 0.4 to 2.5 for the $\Sigma13b$ GB, altering the local composition relative to predictions based on the Langmuir-McLean isotherm. These findings underscore the critical role of secondary GB dislocations not just as a topological necessity to accommodate GB crystallographic discontinuities but also as deep solute traps, massively modulating solute segregation by up to a factor of two. This enhances our understanding of GB decoration, offering opportunities for the design of advanced alloys, such as altering the texture or changing the grain shape to modulate secondary GB dislocations for specific segregation states and properties.

## Methods
### Sample preparation
We deposited the model alloy Fe-1at%W thin film sample in a PVD cluster (BESTEC GmbH, Berlin, Germany). The synthesis process involved co-sputtering a pure Fe target (99.995%, Mateck, Germany) in a direct current cathode with a power of 130 W, and a W target (99.95%, Kurt J. Lesker, USA) in a radio frequency cathode at 28 W, resulting in a total thickness of approximately 2000 nm. Prior to sputtering, the chamber was pumped to a base pressure of $4.0 \times 10^{-8}$ mbar. The Fe-W alloy thin films were then deposited at a pressure of $5.0 \times 10^{-3}$ mbar and an Ar flux of 40 sccm on smooth substrates of single crystalline silicon [100] wafers with a 1.5 µm thermal $SiO_2$ diffusion and reaction barrier layer (see Supplementary Fig. 1a), which was placed 110 mm away from the sputtering targets. The specimens were held at 500 °C for 240 min to facilitate solute segregation at GBs.

We prepared the needle-shaped specimen for correlative 4DSTEM tomography and APT characterization using the lift-out and annular milling technique developed by Thompson et al.[64,65]. The preparation of the specimen was conducted in a plasma-FIB (PFIB) instrument (FEI Helios PFIB) equipped with an Xe-ion source, which results in a low penetration depth that minimizes ion implantation and amorphization in the specimen. The lift-out wedges were directly mounted onto an APT-compatible cylindrical Cu post, ready for ±90° rotation along the tilting axis in the Fischione Model 2050 On-Axis Rotation Tomography Holder in the TEM. During the lift-out procedure, we carefully aligned the rotation axis of the sample with that of the Cu post to match the tilt axis of the tomography holder. A final PFIB milling condition of 10 pA at 5 keV was applied to shape the needle-shaped specimen to a top diameter of 100 nm (see Supplementary Fig. 1e), while ensuring a clean surface.

### Imaging secondary grain boundary dislocations
To observe secondary GB dislocations, we set up multiple two-beam conditions to acquire bright-field, dark-field, and WBDF images. A Mel-

Build HATA tomography holder (with an $\alpha$ tilt range of −60° to +60°) was employed for the observations on a JEM-2200FS microscope (JEOL Ltd.) operated at 200 kV. Low-tilt observations (<15°) were conducted using the Image Cs-corrected Titan Themis 60−300 (Thermo Fisher Scientific), operated at 300 kV. High-resolution STEM imaging of the Σ5 GB was carried out using a Probe Cs-corrected FEI Titan Themis 60−300 (Thermo Fisher Scientific), operated at 300 kV, which enabled visualization of the atomic structure near the surface termination of a secondary GB dislocation.

### Four-dimensional scanning transmission electron microscopy tomography

We conducted the 4DSTEM tomography characterization using a JEM-2200FS (JEOL Ltd.) microscope, which was operated at an accelerating voltage of 200 kV and equipped with a 10 μm condenser (CL1) aperture and a 50 eV energy filter. This 4DSTEM tomography method relies on virtual dark-field images reconstructed from 4DSTEM datasets acquired at different tilts[31,32]. A schematic of the 4DSTEM tomography characterization techniques is presented in Supplementary Fig. 8a. We collected 17 4DSTEM scans at tilts ranging from −80° to +80° with a tilt step increment of 10°. During the scan, a quasi-parallel electron probe, approximately 2 nm in diameter, traverses a selected area of 100 nm × 300 nm in a line-by-line manner with a 1 nm step, capturing nanobeam diffraction patterns at each probe position using a 4k × 4k CMOS detector (TemCam-XF416-TVIPS) with a pixel dwell time of 41 ms.

The phase and orientation of the grains in the 4DSTEM datasets were determined automatically using the multi-index algorithm, which retrieves information about overlapping grains via the ASTAR software package[66,67] (see Supplementary Fig. 7). This crystallographic information serves as a guideline for (1) precisely refining the tilt angles among different tilts; and (2) automatically generating virtual dark-field images for each grain in every 4DSTEM scan using the frozen template algorithm[31,32,67]. We conducted the coarse alignment of the virtual dark-field images manually in Tomviz[68], followed by fine alignment using the PyCorrectedEmissionCT (corrct) package[69]. These well-aligned virtual dark-field images of each grain were used for 3D tomographic reconstruction through the Simultaneous Iterative Reconstruction Technique (SIRT) algorithm, which included 15 iterations combined with a non-negative minimum constraint to promote physical solutions. For accurate determination of GB locations in volumes of overlapping grains, a max-pooling type approach is applied, which assigns the highest intensity volumes to the corresponding grains. We then employed the marching cubes algorithm to generate normals for the planes corresponding to the two grains adjacent to each GB, thus providing distinct plane definitions for both[70]. The spatial resolution of the 3D imaging is constrained by the number of tilt images and feature size, following the Crowther criterion[71], which estimates a 3D spatial resolution of approximately 10 nm. Accordingly, in this study, features smaller than 10 nm could not be resolved. Given that spatial resolution errors may be introduced during 4DSTEM data collection, alignment, and reconstruction, as well as through meshing and the inevitable smoothing algorithms, we focus on capturing the microscopic curvature and orientation of the GB surface as a reference, rather than precisely matching the orientation of every mesh element to the corresponding GB planes. We used Blender[72] to animate the method and visualize the reconstructed volume, as demonstrated in Supplementary Movies 1−3.

### Correlative atom probe tomography

We analyzed the chemistry of the same needle-shaped specimen using APT after acquiring the 4DSTEM datasets. Prior to loading the specimen into the APT analysis chamber, it was cleaned with low-kV Ar ions using a Gatan PECS Model 682 system at 2 kV and 32 μÅ to remove hydrocarbon layers accumulated during the TEM measurements. We

then conducted the APT measurements using a LEAP 5000XS instrument (Cameca Instruments) operated with a specimen temperature of 70 K and a laser pulse energy of 100 pJ at a pulse repetition rate of 200 kHz for a 2% ions per pulse detection rate. As shown in Fig. 2d and Supplementary Fig. 8b, we successfully collected 300 million ions from this correlative specimen, a critical volume that enables the simultaneous investigation of multiple GBs. Data reconstruction was performed using the AP Suite 6.3 software package, following a calibration procedure to achieve the correct image compression factor and k-factor, which are essential for accurately shaping and spacing the lattice in the reconstructed volume[73].

We employed the APT_GB software[41] to analyze the in-plane chemical distribution of W atoms at the GBs. With the pre-trained convolutional neural network, we automatically identified the locations of GB planes from the APT dataset, which were represented as triangular meshes with an average unit size of approximately 7 nm². We generated integral profiles across the GBs to quantify the IE at each vertex on the mesh, with examples shown in (Fig. 4a). The resulting IE map is shown in Fig. 4b.

### Displacement shift complete lattice of the Σ13b grain boundary

The dichromatic pattern of the DSC lattice of the Σ13b GB (see Supplementary Fig. 15) and the analysis of Burgers vectors for secondary GB dislocations were obtained using the DSC lattice through O-lattice theory[74], utilizing an in-house developed open-source GB code[75].

### Stress field simulation

Misfit dislocations on GBs can induce a stress field that would potentially change the segregation behavior of solute atoms in the alloy system. Such stress field was studied in previous work[59,76], where a Stroh formalism is used to obtain the corresponding periodic stress field generated by semicoherent boundary and the required misfit dislocations. In essence, we solve the mechanical equilibrium equation written as

$$\sigma_{ij,j}(x_1, x_2) = C_{ijkl}\, u_{k,jl}(x_1, x_2) = 0 \tag{3}$$

in both upper and lower grain with the critical interface condition expressed as

$$[[\sigma_{2k}(x_1, x_2 = 0)]] = 0$$
$$[[u_k(x_1, x_2 = 0)]] = -\sum_{n=1}^{\infty} \frac{1}{\pi n} \sin\left(\frac{2\pi n}{d} x_1\right) b_k \tag{4}$$

where $[[f]]$ denotes the jump function across the interface (interface normal is $x_2$ axis), $d$ is the dislocation spacing, $b_k$ is dislocation Burgers vector. This condition ensures that traction is in equilibrium across the interface and the relative displacement is consistent with our interfacial dislocation pattern.

To solve the equilibrium (3) on a periodic domain, Fourier series are used to write displacement field.

$$u_k(x_1, x_2) = 2\,\mathrm{Re}\sum_{n=1}^{+\infty} e^{i\frac{2\pi n}{d} x_1} \tilde{u}_k(x_2) \tag{5}$$

where $\tilde{u}_k(x_2)$ is the Fourier coefficient depending on position $x_2$. Substituting the displacement field into mechanical equilibrium together with further manipulation, the system boils down to a sextic equation which is the same with the one in a common Stroh formalism. To satisfy interfacial boundary condition, six complex scaling parameters can be assumed, which will then transform (4) to a set of six independent linear equation. Once the six complex scaling parameters are solved, the stress field can be obtained by calculating the strain with the solved displacement field and multiplying with the elastic constant in each grain (see Supplementary Figs. 13 and 14). The

corresponding Stroh formalism and its intermediate variables are solved via `atomman` package[77].

Area size: $L = 48 \times 48 \, \text{nm}^2$; Number of grid points: $500 \times 500$; Upper grain Euler angles: [41, 54.7, 45]; Lower grain Euler angles: [8, 54.7, 45]; Elastic constants of BCC Fe[78]: $C_{11} = 257.7 \times 10^9$ Pa, $C_{12} = 144.0 \times 10^9$ Pa, $C_{44} = 94.9 \times 10^9$ Pa; Poisson's ratio: $v = 0.3$; The core size of dislocations is defined as 2 Burgers vectors.

**Secondary grain boundary dislocation density calculation.** As discussed for the Σ13*b* GB, the Burgers vector density required to accommodate the interfacial dislocation content can be obtained using the Frank-Bilby equation[62,63]:

$$\mathbf{B} = (\mathbf{I} - \mathbf{P}^{-1})\,\mathbf{t} \qquad (6)$$

where **B** is the Burgers vector density at the GB, **P** is the total misorientation matrix, and **t** is a unit vector lying in the GB plane (orthogonal to the tilt axis).

For the misorientation deviation associated with the Σ13*b* GB, we define **P** as:

$$\mathbf{P} = \mathbf{U}\,\mathbf{R}(\theta)\,\mathbf{U}^{-1} \qquad (7)$$

where $\mathbf{R}(\theta)$ is the rotation matrix corresponding to the misorientation deviation $\theta$ from the ideal CSL GB, and **U** is the transformation matrix from the crystal frame to the global reference frame.

To describe the inclination of the interface, the vector **t** is constructed as:

$$\mathbf{t} = \mathbf{U}\,\mathbf{R}(\phi)\,\mathbf{U}^{-1}\,\mathbf{t}_0 \qquad (8)$$

where $\mathbf{t}_0$ is the reference direction corresponding to zero inclination (i.e., a symmetric tilt GB), and $\mathbf{R}(\phi)$ is the rotation matrix defined by the inclination $\phi$.

Due to the non-orthogonality of the DSC lattice vectors in the Σ13*b* GB, the Burgers vector **B** is decomposed into two DSC components, as illustrated in Supplementary Fig. 15, with corresponding density coefficients $\alpha$ and $\beta$. The matrix **A**, composed of the two DSC vectors (DSC-a and DSC-b). The decomposition is written as:

$$\begin{bmatrix} \alpha \\ \beta \end{bmatrix} = (\mathbf{A}^{\mathsf{T}}\mathbf{A})^{-1}\mathbf{A}^{\mathsf{T}}\mathbf{B} \qquad (9)$$

where $\mathbf{A} = [\text{DSC-a}, \text{DSC-b}]$.

**Reporting summary**
Further information on research design is available in the Nature Portfolio Reporting Summary linked to this article.

## Data availability
The data generated in this study have been deposited in the public community repository Figshare: https://doi.org/10.6084/m9.figshare.29149364[79]. Source data are provided with this paper.

## Code availability
The Python code used for the atom probe analysis in this study is available on GitHub: https://github.com/RhettZhou/APT_GB and archived on Zenodo[80].

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

## Acknowledgements

The authors acknowledge Prof. Dr. Christian Liebscher, Dr. Michael Herbig and Dr. Saurabh Mohan Das for the discussion on transmission electron microscopy results. X.Z., W.L., and E.R. acknowledge funding by the German Research Foundation (DFG) through the project HE 7225/11-1. X.C. and B.G. gratefully acknowledge the Collaborative Research Centre/Transregio (CRC/TRR) 270 HoMMage-Z01 project funded by the DFG.

## Author contributions

X.Z., W.L., E.R., X.C. and B.G. secured funding. X.Z. conceived of the presented idea and supervised the project. X.C. conducted the experimental study and data analysis. W.G., P.H., G.D., W.L., E.R., and X.Z. contributed to the electron microscopy data analysis. Y.H. and Y.G. performed the analytic simulations. B.G. and X.Z. contributed to the atom probe tomography data analysis. X.C., X.Z., and D.R. wrote the original paper. All the authors revised the paper.

## Funding

## Competing interests

The authors declare no competing interests.
