## [Transparent Peer Review file · Nature Communications]

Secondary Grain Boundary Dislocations Alter Segregation Energy Spectra

Corresponding Author: Dr Xuyang Zhou

Version 0:

Reviewer comments:

Reviewer #1

(Remarks to the Author)

This study aims to elucidate and quantify the segregation behavior at grain boundaries (GBs), particularly focusing on the role of secondary GB dislocations in the W-doped iron, which was experimentally accomplished by using advanced 3D orientation (4D-STEM) and composition (APT) tomographic reconstruction techniques. Overall, the manuscript is noteworthy for its conciseness and logical structure. However, it is crucial to point out that this work does not exhibit significant novelty or scientific impact. Extensive computational and experimental research utilizing both diverse and analogous techniques has shown consistent solute segregation behaviors at GBs [Phys. Rev. Lett. 121 (2018) 015702; Nat. Commun. 14 (2023) 3535; Comput. Mater. Sci. 232 (2024) 12596]. While the primary focus may not be on secondary GB dislocations, it can be easy to anticipate that these dislocations, as a type of GB defect, will also lead to or even enhance solute segregation. Furthermore, it appears impractical to adjust the properties of the alloy through the modulation of secondary GB dislocations. Whether the highly local chemical fluctuation of the solute atom has any (negative) effect on the global properties should be carefully evaluated. Therefore, it is not recommended to be published in the prestigious journal of Nature Communications. Here are some additional comments:

1. Secondary GB dislocations identified in the virtual DF reconstruction are obscure and debatable. Considering all experimental conditions, it is possible to clearly image these dislocations in the diffraction contrast TEM method, as demonstrated by most research on secondary GB dislocations. Alternatively, atomic-resolution or low-mag HAADF and LAADF imaging techniques are particularly effective for discerning the contrasts of the edge-on secondary GB dislocations and steps. This approach has proven viable for analyzing the GB segregations in the APT tips.
2. Data regarding segregation at GB facets are missing (Supplementary Fig. 6c-e).
3. The peaks in the W-profiles (Fig. 2e and i) can be understood by the dislocation at GBs. How are the lowest points and slopes in these profiles interpreted?
4. The effects of deviation degrees of the CSL misorientations on the mean segregation energy should be discussed. Why can $\Delta S_{\text{seg}}^{\text{XS}}$ be neglected?

Reviewer #2

(Remarks to the Author)

The manuscript reports detailed and spatially localized segregation to grain boundaries and how secondary dislocations in particular influence the segregation of W in Fe. On one hand, the findings are not surprising, but on the other hand the detail is unprecedented and this merits publication. I should also mention that the segregation spectra in Figs. 3e-g are completely novel and confirm earlier computational predictions; this is also of great value to the grain boundary community. While I think this paper is both valuable and important, I also have several suggestions for possible improvement.

1. Throughout the paper, beginning in the abstract, it is stated or implied that the secondary grain boundary dislocations (SGBDs) accommodate grain boundary curvature and therefore segregation is related to curvature. While this is not inaccurate, it masks the length scale dependence. In fact, SGBDs appear because of a change in grain boundary plane orientation and this is a scale independent phenomenon, depending only on the change in the angle. Curvature (units of inverse length) depends on the length scale. If a GB bends through 30° of arc, the same SGBDs will be introduced, independent of size. If each SGBD has the same power to trap W, then the grain boundary excess will also depend on the

size. My recommendation is to describe the SGBDs as arising from changes in grain boundary plane orientation; it then follows that when this happens over a small length scale, the effect on the excess is greater. I am sure the authors could write a simple expression for the excess associated with the SGBDs as a function of grain size. The high density of SGBDs at small length-scales might partially explain the excess solubility of nanocrystalline alloys compared to microcrystalline alloys, without resorting to the "negative grain boundary energy" discussed by others.

2. To me, the description on the grain boundary (GB1) in the text and what is presented in Fig. 2b are not consistent. It is described as a σ_5 symmetric tilt grain boundary (STGB). The first issue is that STGBs are isolated points in the 5D space, while this boundary clearly bends through many degrees of arc (the majority colors range from green to blue, suggesting something like 40° of arc). Only selected points could possibly be STGBs. The second issue is that tilt boundaries for σ_5 have indices (0kl). On the legend, this is the zone from (001) and (101), which do not appear to be prominent colors in Fig. 2b. Finally, the STGBs for σ_5 are (031) and (012) and the colors associated with these orientations are not prominent in Fig. 2b. This might be resolved by showing the orientation of [100] disorientation axis in the figure.

3. The description of Figs. 3f,g say that the black arrows, associated with the STGBs, show significant deviations from the Skew normal distribution. This is not at all obvious to me. As a fractional difference between the data and the curve, I see places that deviate more. How do we know these deviations are not simply experimental uncertainty?

Small comments

4. p. 2. "planar defects" I recommend two-dimensional defects.

5. p. 2. "More specific in" I recommend more specifically in ...

6. p. 2 "Real GB structures invariably break down into facets and regular patterns formed by secondary GB dislocations, to accommodate topological constraints [1, 2]. These curved surfaces"

Are they facets or are they curved? They can't be both singular and rough.

7. Fig. 1 and 2. How is the APT tip aligned with the film growth direction?

Reviewer #3

(Remarks to the Author)

This manuscript has conducted an excellent job on quantifying the relationship between secondary GB dislocations and their segregation energy spectra within a model Fe-W alloy. The authors claim that secondary GB dislocations can have an additional and, in some cases, even a much stronger effect on GB segregation than defect-free GBs. This is an interesting topic and the experimental data can well support the main conclusion of this paper. For the benefit of the readers, this manuscript needs a minor revision before acceptance for publication.

They are given below:

1. In Figures 3 B and C, the authors have quantitatively shown the interfacial solute excess of W segregation and the corresponding segregation energy across different grain boundaries. Nevertheless, the focus of this work is to highlight the effect of secondary GB dislocations on interfacial segregation, as driven by the GB curvatures. Thus, it is recommended that the grain boundary curvature, secondary dislocation density map, interfacial solute excess of W and the calculated segregation energy from the same region should be present together.

2. There remains a significant difference between the experimental GB image and the theoretical model. If possible, the AC-STEM is suggested to characterize the real GB structure at the atomic scale, as this would provide a more intuitive understanding of the atomic arrangement of secondary grain boundary dislocations.

3. I am interested about the spatial resolution error associated with the orientation maps of the local normal to the GB plane. Specifically, I would like to know whether such high spatial resolution can be achieved and, if so, how it is accomplished. Please provide a detailed explanation in the revised text.

Version 1:

Reviewer comments:

Reviewer #1

(Remarks to the Author)

The major concerns have been addressed by the authors.
This work can be considered for being published now.

Reviewer #2

(Remarks to the Author)

The authors have addressed the comments effectively. Publication is recommended.

Reviewer #3

(Remarks to the Author)

Author response to reviewer comments

REVIEWER COMMENTS

Reviewer #1: This study aims to elucidate and quantify the segregation behavior at grain boundaries (GBs), particularly focusing on the role of secondary GB dislocations in the W-doped iron, which was experimentally accomplished by using advanced 3D orientation (4D-STEM) and composition (APT) tomographic reconstruction techniques. Overall, the manuscript is noteworthy for its conciseness and logical structure. However, it is crucial to point out that this work does not exhibit significant novelty or scientific impact. Extensive computational and experimental research utilizing both diverse and analogous techniques has shown consistent solute segregation behaviors at GBs [Phys. Rev. Lett. 121 (2018) 015702; Nat. Commun. 14 (2023) 3535; Comput. Mater. Sci. 232 (2024) 112596]. While the primary focus may not be on secondary GB dislocations, it can be easy to anticipate that these dislocations, as a type of GB defect, will also lead to or even enhance solute segregation. Furthermore, it appears impractical to adjust the properties of the alloy through the modulation of secondary GB dislocations. Whether the highly local chemical fluctuation of the solute atom has any (negative) effect on the global properties should be carefully evaluated. Therefore, it is not recommended to be published in the prestigious journal of Nature Communications.

Response: We sincerely appreciate the positive remarks regarding the conciseness and logical structure of our manuscript. We are also grateful for the insightful and constructive comments of the reviewer, which have helped us clarify the novelty and scientific significance of this study.

As pointed out by the reviewer, solute segregation at grain boundaries (GBs) has been intensively explored since the mid-twentieth century, from the early studies, e.g., Suzuki [J. Phys. Soc. Jpn. 17 (1962) 322] to the recent studies cited by the reviewer. Because segregation governs many global properties—such as ductility, impact strength, and fracture toughness—this topic continues to receive considerable attention. Nevertheless, a complete understanding of the fundamental solute–defect interactions remains elusive owing to the complex GB structure and experimental limitations. In addition to the five kinematic degrees of freedom intrinsic to a GB, secondary GB dislocations introduce an additional level of structural complexity. Additionally, although it is beyond the scope of the paper to experimentally study the influence of highly localized chemical fluctuations of the solute atoms on the global properties, we have included an outlook discussing how local chemical fluctuations of solute atoms might affect global properties.

The reviewer correctly notes that secondary GB dislocations, as defects of the GB, can promote solute segregation. Beyond confirming this qualitative expectation, our contribution quantifies the effect of secondary GB dislocations on the segregation-energy landscape and elucidates the underlying interaction mechanism. In the revised manuscript, we have included a more detailed analysis of secondary GB dislocations. By combining high-resolution transmission electron

microscopy (HRTEM), four-dimensional scanning transmission electron microscopy (4D-STEM) tomography and atom probe tomography (APT), we obtained the three-dimensional (3D) crystallographic information of GBs, imaged and analyzed secondary GB dislocations, including determining their Burgers vectors via HRTEM, and employed interfacial excess mapping to extract 3D, spatially resolved segregation energy distributions, an experimental capability that, to our knowledge, has not been demonstrated previously.

The resulting segregation spectra (Figs. 4e–g) provide near-atomic-scale evidence for solute partitioning at individual secondary GB dislocations. Such a fundamental achievement can not only provide support for existing literature, including the theoretical work mentioned by the reviewer [Comput. Mater. Sci. 232 (2024) 112596], but also offer further insights for theory development, for example, by supplying crystallographic information on secondary GB dislocations together with the corresponding segregation spectrum. In addition, we have added a dedicated discussion comparing our results with those cited by the reviewer, highlighting the unique contributions and distinguishing features of our approach.

We have revised the phrasing 'adjust the properties of the alloy through the modulation of secondary GB dislocations' to: 'This knowledge is essential for quantifying the interactions between secondary GB dislocations and alloying elements.' Such understanding opens new avenues to exploit segregation effects, enabling precise tailoring of material properties.

In the revised paper, we further conducted a systematic analysis and simulation of the effects of GB inclination and misorientation on the segregation patterns (Fig. 5f–i), which we believe enhances the depth of the study and provides a more comprehensive explanation of the observed phenomena. We trust that the revised manuscript addresses the concerns of the reviewer and highlights the originality and relevance of our findings to the broader materials-science community.

Changes made in response to the comment:

Page 2, line 51-57, added: “Reported atomic-scale investigations clarify the role of GB defects in solute segregation. For instance, Hu et al. showed that GB dislocations and steps along Pt twin boundaries distorted local elastic fields and altered Au segregation [15–17], whereas Liebscher et al. observed enrichment of C, Fe, and N at specific facet junctions of faceted, tilted Si GBs [18]. These observations suggest that GB defect-mediated segregation is widespread, yet its magnitude across diverse GB defect structures remains to be rigorously quantified experimentally.”

Page 2, line 60-62, modified: “Systematically characterizing these complex solute–GB interactions with highest spatial resolution and chemical sensitivity is critical for understanding GB decoration and unlocking atomic-scale engineering of alloys [20].”

Page 3, line 95-101, added: “This knowledge is essential for quantifying the interactions between secondary GB dislocations and alloying elements, opening design opportunities for atomic engineering of future materials. For instance, tailoring texture to modulate secondary GB dislocations can be used to further adjust solute segregation through localized elastic interactions and ultimately enhance ductility, impact strength, and fracture toughness.”

Fig. 1. Secondary GB dislocations in the Fe-1 at.% W specimen... **g** Bright-field image of a $\Sigma 3$ GB with a 2° deviation from the ideal misorientation. **h** Corresponding dark-field image of the $\Sigma 3$ GB with $g = [0\bar{1}1]$, revealing secondary GB dislocations. **i** Bright-field image of a $\Sigma 5$ GB that deviates by 2.5° from the ideal misorientation. **j** Dark-field images of the $\Sigma 5$ GB acquired at various α tilt angles. Scale bar: 50 nm. **k** Magnified view of the area at 0° tilt angle, with semi-transparent red dots highlighting the atomic column configuration near the GB. See additional details in Supplementary Fig. 5. **l** The region in **k** overlaid with a displacement shift complete (DSC) lattice and a Frank circuit, illustrating the presence of a secondary GB dislocation characterized by the Burgers vector **b**.

Fig. 5. Quantifying elastic energy for the $\Sigma 13b$ GB with dislocations using linear anisotropic elasticity theory... **f** Maximum change in segregation energy (Δ Seg. Energy) caused by DSC-a or DSC-b dislocations as a function of inclination ϕ . The blue and red circles indicate the experimentally relevant inclinations. **g** Dislocation density maps for DSC-a and DSC-b types as a function of inclination ϕ and misorientation deviation θ from the ideal $\Sigma 13b$ GB (rotation axis $[111]$), derived using the Frank–Bilby equation. **h** Dislocation densities for DSC-a and DSC-b as a function of misorientation deviation θ at a fixed inclination of $\phi = 15.6^\circ$. **i** Calculated segregation energy profiles for the $\Sigma 13b$ GB with a misorientation deviation of $\theta = 0.6^\circ$, showing contributions from DSC-a, DSC-b secondary GB dislocations, and their combined effect.

Here are some additional comments:

Reviewer #1 Comment 1: Secondary GB dislocations identified in the virtual DF reconstruction are obscure and debatable. Considering all experimental conditions, it is possible to clearly image these dislocations in the diffraction contrast TEM method, as demonstrated by most research on secondary GB dislocations. Alternatively, atomic-resolution or low-mag HAADF and LAADF imaging techniques are particularly effective for discerning the contrasts of the edge-on secondary GB dislocations and steps. This approach has proven viable for analyzing the GB segregations in the APT tips.

Response: We appreciate the reviewer for raising the concern that the secondary GB dislocations identified in our virtual dark-field (DF) reconstruction may be open to further verification. We fully understand this concern and, guided by the suggestion of the reviewer, performed additional targeted experiments.

Specifically, in addition to the diffraction-contrast TEM image shown in the original Fig. 1e (that is, the weak-beam dark-field, WBDF, image), we carried out further work that directly compares WBDF images with virtual DF reconstructions from 4DSTEM for the same GB (see the newly added Supplementary Fig. 3 and Supplementary Fig. 4 in the revised manuscript). Both methods yield consistent diffraction contrast and dislocation imaging, thereby making the secondary GB dislocations readily visible. It is worth noting that thickness fringes are also observed, which can vary with acceleration voltage (e.g., 200 kV vs 300 kV). Furthermore, diffraction-contrast images were produced from sixteen individual reflections within the $[001]$ zone-axis diffraction pattern using both methods (Supplementary Fig. 4), further demonstrating the robustness of the virtual DF approach to image dislocations by 4DSTEM.

Following the suggestion of the reviewer, we also acquired low-magnification HAADF and atomic-resolution low-angle ADF (LAADF) images from the same GB region (Supplementary Fig. 2). These images resolve the secondary GB dislocations arrays even in the low resolution HAADF image and their local atomic structures, corroborating the observations in Fig. 1d–f. To visualize the atomic-scale contrast of edge-on secondary GB dislocations and the associated steps along a $\Sigma 5$ boundary, we have added Fig. 1i–l and Supplementary Fig. 5. These HAADF images clearly display the edge-on secondary GB dislocations.

Beyond our own data, the existing literature has established virtual DF reconstruction as a powerful tool for dislocation imaging. For example, Pham *et al.* [Nature Materials 24 (2025) 682] resolved individual dislocations in organic molecular crystals by virtual DF imaging, illustrating the capability of 4DSTEM for detailed dislocation analysis even in complex materials.

Finally, virtual DF offers key advantages over conventional diffraction-contrast TEM. Traditional double-tilt TEM and atomic-resolution STEM, although effective, face practical limitations when applied to polycrystalline or randomly oriented nanocrystalline specimens, particularly when the specimen holder cannot reach large α and β tilt angles (limited by the objective aperture). In contrast, 4DSTEM tomography collects diffraction patterns over a broad orientation range, enabling post-acquisition DF reconstructions for any visible reflection and revealing secondary GB dislocation contrast across the GB. Virtual DF can be generated for every recorded diffraction spot, providing a comprehensive view in both real and reciprocal space that conventional DF imaging cannot match.

We trust that these additional data and clarifications address the concern of the reviewer and further strengthen the validity and rigor of our methods.

Changes made in response to the comment:

To better present the additional characterization of the secondary GB dislocations, the original Fig. 1 has been divided into Fig. 1 (showing the new results) and Fig. 2. Newly added images appear in Fig. 1g–l (refer to our first response to Reviewer #1). We have also added Supplementary Figs. 2–6. Supplementary Figs. 3 and 4 compare diffraction-contrast images with virtual DF imaging, while Supplementary Figs. 2 and 5 provide atomic-resolution and low-magnification HAADF and LAADF images. Supplementary Fig. 6 shows a perfect Frank circuit to assist in measuring the Burgers vector associated with the secondary GB dislocation in Fig. 1l.

Page 28-32, added Supplementary Figs. 2 - 6:

Supplementary Fig. 2. High-resolution characterization of secondary GB dislocations on the Σ_{11} GB, shown in Fig. 1d. **a** Low-magnification low-angle annular dark-field (LAADF) image of the Σ_{11} GB, highlighting periodic contrast variations associated with the secondary GB dislocation, with boxed areas indicating regions of interest for further analysis. **b & c** High-angle annular dark-field (HAADF) images of the red and blue dashed boxes shown in **a**, showing secondary GB dislocations along the GB plane. Variations in contrast indicate possible local compositional undulations. **d** Schematic illustration of the secondary GB dislocation network extracted from **c**. **e** LAADF image from the red boxed region shown in **c**, revealing the atomic structure near the termination of a secondary GB dislocation. The red-boxed region highlights where a secondary GB dislocation terminates. **f** Inverse fast Fourier transform (iFFT)-filtered image of the same region, showing that the presence of a secondary GB dislocation alters the local periodicity and induces lattice distortion. For this GB, Frank circuit analysis could not be performed to quantify the Burgers vector of this secondary GB dislocation. The blue dashed regions indicate the potential presence of other bulk dislocations.

Supplementary Fig. 3. Comparison between TEM WBDF and 4DSTEM dark-field imaging for secondary GB dislocations. **a** Selected area electron diffraction pattern of the bottom grain in Supplementary Fig. 4 with labeled diffraction spots used for WBDF imaging. **b & c** WBDF images acquired under the two-beam condition using diffraction spots **b** and **c** in **a**, respectively, revealing the periodic contrast associated with secondary GB dislocations. **d** Converged beam electron diffraction pattern of the bottom grain in Supplementary Fig. 4, showing the spots selected for virtual dark-field image reconstruction. **e & f** Virtual dark-field images reconstructed using diffraction spots **e** and **f** in **d**, respectively. The 4DSTEM dataset obtained under the two-beam condition of spot **f** provides comparable dislocation contrast to the conventional WBDF images. Scale bar: 50 nm.

Supplementary Fig. 4. Comparison between conventional dark-field and virtual dark-field imaging of the same GB, including two associated image series. a Bright-field TEM image of the $\Sigma 11$ GB shown in Fig. 1d with the dashed red box indicating the area used for the image series in e. **b** Selected area electron diffraction pattern corresponding to the bottom grain in the dashed red box in a. **c** Bright-field STEM image of the $\Sigma 11$ GB region, with the dashed blue box showing the area analyzed in f. **d** Converged beam electron diffraction pattern from the bottom grain in the dashed blue box in c. **e** Conventional dark-field images corresponding to individual diffraction spots labeled 1 to 16 in b, acquired across the dashed red boxed region in a, revealing contrast variations along the GB. **f** Virtual dark-field image series based on individual diffraction spots labeled 1 to 16 in d, obtained from the dashed blue boxed region in c.

Supplementary Fig. 5. Atomic-scale imaging of a GB region containing secondary GB dislocations. **a** Low-magnification STEM image showing the GB region of interest. The dashed red box highlights the region selected for high-resolution imaging, as shown in **b**. Inset: a larger field of view indicating the GB location. **b** High-resolution HAADF image of the selected region in **a**. **c** Magnified view of the region indicated by the dashed blue box in **b**. **d** Further magnification of the dashed red boxed region in **c**, clearly resolving atomic columns and revealing structural distortion at the GB. **e** Fast Fourier transform (FFT) of the high-resolution STEM image in **d**. Red circles indicate the selected reflections used for inverse filtering. **f** iFFT image reconstructed using the selected reflections from **e**. The pink dashed box indicates the region used in Fig. 1k for the secondary GB dislocation analysis.

Supplementary Fig. 6. The defect-free reference Frank circuit of the $\Sigma 5$ GB corresponding to the image shown in Fig. 11. The circles represent atomic positions of the upper grain (gray) and the lower grain (pink). CSL: coincident site lattice. DSC: displacement shift complete.

Page 3-5, line 115-137, added: “Additional information from high-angle annular dark-field (HAADF) STEM imaging (see Supplementary Fig. 2) further reveals the secondary GB dislocation network and the associated local lattice distortions.

To provide an independent check, we acquired 4DSTEM data and generated virtual dark-field images (see Supplementary Figs. 3 and 4), which reproduce the contrast observed in conventional WBDF images. This demonstrates that secondary GB dislocations are consistently detected by both techniques and confirms their role in producing the observed intensity modulation.

Figures 1g–h show a similar analysis for a $\Sigma 3$ GB with a 2° deviation, where secondary GB dislocations are also observed under two-beam conditions with $g = [0\bar{1}1]$. The GB can be decomposed into a series of planar segments, each characterized by a distinct density of secondary GB dislocations. Figure 1i presents the periodic contrast features arising from secondary GB dislocations in a $\Sigma 5$ GB exhibiting a 2.5° deviation, while Fig. 1j shows these features under varying tilt angles. Figure 1k provides atomic-resolution images of the selected region in Fig. 1j, capturing the termination point of a secondary GB dislocation in the $\Sigma 5$ GB. The atomic columns around the GB are highlighted with red markers. We overlay the observed structure with a displacement shift complete (DSC) lattice and construct a Frank circuit, as shown in Fig. 11 (also see Supplementary Fig. 6 for the reference circuit), to determine the projected Burgers vector along the beam direction associated with the secondary GB dislocation.”

Page 15, line 402-411, added: “To observe secondary GB dislocations, we set up multiple two-beam conditions to acquire bright-field, dark-field, and WBDF images. A Mel-Build HATA tomography holder (with an α tilt range of -60° to 60°) was employed for the observations on a JEM-2200FS microscope (JEOL Ltd.) operated at 200 kV. Low-tilt observations ($<15^\circ$) were conducted using the Image Cs-corrected Titan Themis 60-300 (Thermo Fisher Scientific), operated at 300 kV. High-resolution STEM imaging of the $\Sigma 5$ GB was carried out using a Probe Cs-corrected FEI Titan Themis 60–300 (Thermo Fisher

Scientific), operated at 300 kV, which enabled visualization of the atomic structure near the surface termination of a secondary GB dislocation.”

Reviewer #1 Comment 2: Data regarding segregation at GB facets are missing (Supplementary Fig. 6c-e).

Response: We thank the reviewer for identifying this mistake in the original submission. The detailed information regarding GB facets should have appeared in Supplementary Fig. 6, but the image for this figure was incorrectly cited. In the revised version, we have updated Supplementary Fig. 6 (now Supplementary Fig. 11 in the revised manuscript) and provided further detailed results regarding segregation at the GB3 facets in Supplementary Fig. 12.

Changes made in response to the comment:

Page 8, line 199-201, modified: “We also observed GB facets that modulate solute segregation at GBs, consistent with previous reports [18, 24, 27, 35]. These were particularly noted at GB3 (see Supplementary Fig. 11 for detailed analysis).”

Page 10, line 240-243, added: “It is noteworthy that this minor deviation can lead to peak splitting into two distinct domains in the segregation energy spectrum, as shown in Supplementary Fig. 12, when readily visible facets appear in GBs (see the facets in Supplementary Fig. 11).”

Page 37, added Supplementary Fig. 11:

Supplementary Fig. 11. Characterization of GB facets and their link to segregation patterns in the Fe-1 at.% W specimen. a The same mapping as in Fig. 3a. **b** The isosurfaces superimposed at 8.0 at.% W on the atom maps of Fe and W highlight the GB

indicated by the red arrow in a. **c** Mapping of the local normal to the GB plane between adjacent grains, from the grain α_3 side and the grain α_4 side, respectively. **d** Side view of the GB within the blue dashed-line frame shown in b, illustrating a staircase GB segregation pattern and indicating GB facets. The superimposed arrays of atoms represent the crystalline lattices for grains α_3 and α_4 . **e** Compositional profile of W along the red arrow in d. The numbers ① and ② in c-e indicate the same positions across different plots.

Page 38, added Supplementary Fig. 12:

Supplementary Fig. 12. Experimental GB segregation energy spectrum for GB3. The fitting curves, overlaid on the histograms, generally follow a skew-normal distribution [37]. For GB3, the skew-normal distribution splits into two distinct domains: Part 1 for ① and Part 2 for ② observed on GB3 (Supplementary Fig. 11).

Reviewer #1 Comment 3: The peaks in the W-profiles (Fig. 2e and i) can be understood by the dislocation at GBs. How are the lowest points and slopes in these profiles interpreted?

Response: We appreciate the insightful question raised by the reviewer regarding the lowest points and slopes in the W-concentration profiles shown in Fig. 2e and 2i (currently Fig. 3e and 3i). The lowest points correspond to W-depleted zones caused by the compressive hydrostatic stress field adjacent to each secondary GB dislocation, consistent with Cottrell-atmosphere theory. The slopes connecting these minima to the neighboring peaks reflect the gradual change in hydrostatic stress from the compressive side to the tensile side of the dislocation line, thereby outlining the segregation-energy gradient responsible for the observed profile shape. Further interpretation of these slopes is provided in the simulation results presented in Fig. 5c–i and the corresponding interpretation text for Fig. 5c–i.

Changes made in response to the comment:

Page 11, line 292-297, added: “The tensile and compressive hydrostatic stress fields adjacent to secondary GB dislocations or GB ledges are the primary reason for the highest and lowest points in the 1D composition profiles plotted in Fig. 3e & i and Figs. 5a & b. The pronounced compositional slope indicates that this undulation is likely caused by stress gradients associated

with secondary GB dislocations, which will be theoretically estimated in the following paragraphs.”

Reviewer #1 Comment 4a: The effects of deviation degrees of the CSL misorientations on the mean segregation energy should be discussed.

Response: We sincerely appreciate the insightful suggestion from the reviewer regarding the influence of coincidence site lattice (CSL) misorientation deviations. Deviations from ideal CSL orientations can alter mean segregation energies by increasing the secondary GB dislocations.

Herbig *et al.* [Phys. Rev. Lett. 112 (2014) 126103], through correlative APT-TEM analysis revealed that even modest deviations from ideal CSL misorientations cause a rapid increase in segregation energy, correlating with enhanced solute capture. This effect stems from the proliferation of secondary GB dislocations upon deviation. These secondary dislocations create high-energy segregation sites, thereby elevating the mean segregation energy. Furthermore, independent atomic-resolution work demonstrates that the absolute amount of segregation results from the interplay between the GB defects and its local strain field [Phys. Rev. Lett. 121 (2018) 015702].

While a comprehensive quantification of deviation angle effects on mean segregation energy requires systematic atomic-scale data for each CSL type at different deviation angles (beyond our current scope), we have strengthened our manuscript by explicitly noting that secondary GB dislocations not only create nanoscale segregation heterogeneity but also enhance mean segregation energy through their dislocation cores.

Inspired by the suggestion from the reviewer, we conducted new quantitative analyses linking segregation energy landscapes to GB kinematic parameters: 1) Misorientation deviation from ideal CSL GB (θ) and 2) Inclination (ϕ) - quantifying local GB plane deviation from symmetric orientations (Fig. 5f-i). Applying Frank–Bilby theory, we demonstrate how dislocation density varies with θ and ϕ (Fig. 5g-h). This variation directly modulates maximum segregation energies at dislocation-containing GB regions, providing mechanistic insight into local segregation behavior.

Changes made in response to the comment:

We have updated Fig. 5 to include the theoretical study of the influence of deviation angles from the CSL misorientation on solute segregation (refer to our first response to Reviewer #1).

Page 12-14, line 322-354, added: “Besides modulating local solute distribution, secondary GB dislocation formation creates high-energy segregation sites at the dislocation cores and introduces additional solute into GBs [18, 45]. Herbig *et al.* demonstrated this through correlative TEM-APT analysis, showing that deviations from ideal CSL misorientations increase mean solute segregation [45]. Our experimentally measured mean solute segregation includes the additional contribution from high-energy segregation sites associated with secondary GB dislocations. This work advances previous studies by quantifying and

interpreting the heterogeneous chemical distribution at defective GBs. In the following, we further strengthen the theoretical understanding by investigating how GB crystallography influences secondary GB dislocation structures and the corresponding in-plane distribution of solute W.

Figure 5f shows the maximum change in segregation energy induced by DSC-a or DSC-b dislocations as a function of the inclination ϕ , where ϕ defines the local deviation of the GB plane from its symmetric orientation. For instance, at $\phi = 0^\circ$, the DSC-b dislocation generates a peak segregation energy increase of approximately +6 kJ/mol. As ϕ increases to $\phi = 90^\circ$, the elastic contribution of the DSC-b dislocation to the segregation energy progressively decreases.

To evaluate how GB crystallography influences segregation, we calculated the densities of DSC-a and DSC-b dislocations as functions of inclination ϕ and misorientation deviation θ from the ideal $\Sigma 13b$ GB (rotation axis [111]), using the Frank–Bilby equation [62, 63] (see Dislocation density calculation), as shown in Fig. 5g. Figure 5h demonstrates that both dislocation densities increase with misorientation deviation θ , with DSC-a and DSC-b exhibiting distinct trends at a fixed inclination of $\phi = 15.6^\circ$, characteristic of the $\Sigma 13b$ GB shown in Fig. 5h inset. Assuming that DSC-a and DSC-b dislocations are each arranged with uniform spacing and positioned independently along the GB, based on the calculated dislocation density, we reconstructed the segregation energy landscape shown in Fig. 5i. The superposition of their elastic fields results in partial cancellation and produces a periodic modulation along the GB. Notably, small secondary spikes appear between the main peaks and troughs, consistent with the fine structure observed experimentally in the blue dashed box in Fig. 5b.”

Page 18-19, line 514-533, added: “As discussed for the $\Sigma 13b$ GB, the Burgers vector density required to accommodate the interfacial dislocation content can be obtained using the Frank–Bilby equation [62, 63]:

$$\mathbf{B} = (\mathbf{I} - \mathbf{P}^{-1}) \mathbf{t} \quad (4)$$

here, \mathbf{B} is the Burgers vector density at the GB, \mathbf{P} is the total misorientation matrix, and \mathbf{t} is a unit vector lying in the GB plane (orthogonal to the tilt axis).

For the misorientation deviation associated with the $\Sigma 13b$ GB, we define \mathbf{P} as:

$$\mathbf{P} = \mathbf{U} \mathbf{R}(\theta) \mathbf{U}^{-1} \quad (5)$$

where $\mathbf{R}(\theta)$ is the rotation matrix corresponding to the misorientation deviation θ from the ideal CSL GB, and \mathbf{U} is the transformation matrix from the crystal frame to the global reference frame.

To describe the inclination of the interface, the vector \mathbf{t} is constructed as:

$$\mathbf{t} = \mathbf{U} \mathbf{R}(\phi) \mathbf{U}^{-1} \mathbf{t}_0 \quad (6)$$

where \mathbf{t}_0 is the reference direction corresponding to zero inclination (i.e., a symmetric tilt GB), and $\mathbf{R}(\phi)$ is the rotation matrix defined by the inclination ϕ . Due to the non-orthogonality of the DSC lattice vectors in the $\Sigma 13b$ GB, the Burgers vector \mathbf{B} is decomposed into two DSC

components, as illustrated in Supplementary Fig. 15, with corresponding density coefficients α and β . The matrix \mathbf{A} , composed of the two DSC vectors (DSC-a and DSC-b). The decomposition is written as:

$$\begin{bmatrix} \alpha \\ \beta \end{bmatrix} = (\mathbf{A}^T \mathbf{A})^{-1} \mathbf{A}^T \mathbf{B} \quad (7)$$

where $\mathbf{A} = [\text{DSC-a}, \text{DSC-b}]$.”

Reviewer #1 Comment 4b: Why can $\Delta S_{\text{seg}}^{\text{XS}}$ be neglected?

Response: We thank the reviewer for this insightful question. A rigorous evaluation of $\Delta S_{\text{seg}}^{\text{XS}}$ demands simultaneous consideration of configurational, vibrational, electronic, and magnetic entropy across the full GB space of polycrystalline materials, a task that remains extremely computationally challenging. Tuchinda and Schuh [npj Comput. Mater. 10 (2024) 72] advanced the field by calculating vibrational contributions for Ag-, Al-, Au-, Cu-, Ni-, Pd-, and Pt-based alloys via the harmonic approximation, but their study does not cover the Fe–W system examined here. Consequently, no reliable quantitative estimate of ΔS_{seg} exists for the present alloy. Their work also reveals an empirical relation $\Delta S_{\text{seg}} = \chi \Delta E_{\text{seg}} + \Delta S_0$, with $\chi \approx 1 \times 10^{-4} \text{ K}^{-1}$ for most systems. At our processing temperature of 500 °C ($\approx 0.43 T_m$ for Fe), the entropy term alters the segregation energy by less than 15 %. Because ΔS_{seg} and ΔE_{seg} vary in the same direction, including ΔS_{seg} would uniformly shift the mean segregation level without affecting the local variations that arise from secondary GB dislocations—the central focus of this work. Neglecting ΔS_{seg} therefore introduces negligible error while keeping the discussion centered on the mechanistic origins of heterogeneous segregation.

Changes made in response to the comment:

Page 11, line 279-283, added: “Computational work by Tuchinda et al. demonstrates that in most alloy systems, the entropy contribution typically alters the segregation energy by less than 15% [50]. We consequently neglect the $\Delta S_{\text{seg}}^{\text{XS}}$ term and primarily attribute the modulation of segregation energy within a GB to the elastic energy resulting from secondary GB dislocations.”

Reviewer #2: The manuscript reports detailed and spatially localized segregation to grain boundaries and how secondary dislocations in particular influence the segregation of W in Fe. On one hand, the findings are not surprising, but on the other hand the detail is unprecedented and this merits publication. I should also mention that the segregation spectra in Figs. 3e-g are completely novel and confirm earlier computational predictions; this is also of great value to the grain boundary community. While I think this paper is both valuable and important, I also have several suggestions for possible improvement.

Response: We greatly appreciate the constructive remarks from the reviewer. The acknowledgement that our segregation spectra are completely novel, that they validate earlier computational predictions, and that they hold significant value for the GB community is particularly encouraging. In the following, we will implement the suggested refinements to further strengthen the manuscript.

Reviewer #2 Comment 1: Throughout the paper, beginning in the abstract, it is stated or implied that the secondary grain boundary dislocations (SGBDs) accommodate grain boundary curvature and therefore segregation is related to curvature. While this is not inaccurate, it masks the length scale dependence. In fact, SGBDs appear because of a change in grain boundary plane orientation and this is a scale independent phenomenon, depending only on the change in the angle. Curvature (units of inverse length) depends on the length scale. If a GB bends through 30° of arc, the same SGBDs will be introduced, independent of size. If each SGBD has the same power to trap W, then the grain boundary excess will also depend on the size. My recommendation is to describe the SGBDs as arising from changes in grain boundary plane orientation; it then follows that when this happens over a small length scale, the effect on the excess is greater. I am sure the authors could write a simple expression for the excess associated with the SGBDs as a function of grain size. The high density of SGBDs at small length-scales might partially explain the excess solubility of nanocrystalline alloys compared to microcrystalline alloys, without resorting to the "negative grain boundary energy" discussed by others.

Response: We thank the reviewer for this highly insightful and constructive comment. The suggestions are indeed very helpful for clarifying the fundamental concepts related to secondary GB dislocations.

As the reviewer correctly notes, the presence of secondary GB dislocations on a curved GB depends on changes in inclination (that is, GB plane orientation) and is independent of scale. We have amended the inaccurate statements accordingly. In both the Abstract and Introduction, we now state that curvature produces discrete variations in GB plane orientation which, together with inclination and misorientation deviations from the perfect CSL GB, are accommodated by topologically required arrays of secondary GB dislocations.

In addition, the reviewer points out that if a GB bends by the same angle, the same secondary GB dislocations will be introduced. We wish to make a small clarification. The bending of a GB at the nanoscale can be treated as being decomposed into many discrete steps. However,

the size and density of these steps cannot be uniformly predicted by a simple expression. Geometrically, a curved boundary can be partitioned into multiple planar segments, but such decomposition is not unique. This is why we cannot provide a simple expression for the excess associated with the secondary GB dislocations as a function of grain size. We believe the reviewer means that the same angular misorientation deviation of a GB from a perfect CSL GB should be compensated by the same number of secondary GB dislocations, or more precisely, by the same Burgers vector density. In the revised version, we have added a theoretical calculation of the dislocation density \mathbf{B} to address this concern. The dislocation density \mathbf{B} is related to the misorientation matrix \mathbf{P} and the GB-plane unit vector \mathbf{t} (related to GB inclination) through the Frank–Bilby equation, $\mathbf{B} = (\mathbf{I} - \mathbf{P}^{-1})\mathbf{t}$. The results and explanation now appear in Fig. 5 and in the section titled “Estimation of the elastic energy contribution for GBs containing secondary GB dislocations.” We also refer the reviewer to our response to Reviewer #1 Comment 4a.

Finally, the secondary GB dislocation density is influenced not only by these topologically required dislocations. Curved GBs often facet to minimize local energy, and facet junctions can introduce secondary GB dislocations of which Burgers vectors can be compensated by other secondary GB dislocations at different facet junctions. Consequently, they do not increase the net Burgers vector density needed to accommodate the deviation from the perfect CSL GB. This effect is more pronounced in smaller grains with higher curvature, which may support the statement of the reviewer that grains with small length scales can have a high density of secondary GB dislocations, leading to the excess solubility observed in nanocrystalline alloys compared to microcrystalline alloys.

We are grateful to the reviewer for prompting this clarification, which we believe improves the rigor and clarity of the manuscript. In accordance with the suggestions of the reviewer, we have revised the manuscript as follows.

Changes made in response to the comment:

Page 1, line 20-23, added: “This leads to discrete variations in the GB plane orientations. Topologically required arrays of secondary GB dislocations accommodate these variations as well as deviations from ideal coincidence site lattice GBs.”

Page 2, line 37-40, added: “At the microscale, GBs accommodate curvature via planar segments of differing inclination, with some segments containing nanoscale steps and a periodic array of secondary GB dislocations, thereby preserving lattice continuity [3–6], as illustrated in Figs. 1c.”

Reviewer #2 Comment 2: To me, the description on the grain boundary (GB1) in the text and what is presented in Fig. 2b are not consistent. It is described as a σ_5 symmetric tilt grain boundary (STGB). The first issue is that STGBs are isolated points in the 5D space, while this boundary clearly bends through many degrees of arc (the majority colors range from green to blue, suggesting something like 40° of arc). Only selected points could possibly be STGBs.

The second issue is that tilt boundaries for sigma_5 have indices (0kl). On the legend, this is the zone from (001) and (101), which do not appear to be prominent colors in Fig. 2b. Finally, the STGBs for sigma_5 are (031) and (012) and the colors associated with these orientations are not prominent in Fig. 2b. This might be resolved by showing the orientation of [100] disorientation axis in the figure.

Response: We thank the reviewer for the careful reading and insightful comments regarding the description of GB1 and its representation in Fig. 3b (Fig. 2b in the original manuscript). We fully understand that our initial description may have caused confusion. The term “ $\Sigma 5$ symmetric tilt GB (STGB)” was intended to refer to the idealized CSL character of the segment at GB1, which approximated the $\Sigma 5$ (210)/[001] configuration. However, as the reviewer correctly pointed out, the actual experimental GB is curved and spans a range of local orientations, deviating from the ideal STGB configuration in a five-dimensional (5D) space. We have revised the description to clarify that the GB1 consists of a range of local orientations rather than representing an ideal $\Sigma 5$ symmetric tilt boundary.

As recommended by the reviewer, we have inserted cubic orientation markers in Fig. 2b—each annotated with the [100] axis to Fig. 3b, as the reviewer suggested, to aid in identifying the local misorientation axis and to further clarify the character of the GB.

We now explicitly note that the GB is geometrically curved, with a range of local orientations, as revealed by the color mapping in Fig. 3b, which indeed spans green to blue hues—indicative of significant angular variation (0~35.3°). We acknowledge that the GB plane reconstruction in 3D using the tomographic technique inevitably contains inherent uncertainty, especially in regions where the interfacial area is small. As we also addressed in our response to Reviewer 3 Comment 3, while we strive for high spatial accuracy, nevertheless, reconstructing every GB plane orientation with absolute precision remains challenging. Therefore, the pronounced curvature in the lower part of GB1 may partially arise from reconstruction-related artifacts. Nevertheless, we believe the major GB remains reliable, both in terms of GB character.

Furthermore, we have updated the figure caption to emphasize that the prominent orientations shown do not exactly correspond to the classic $\Sigma 5$ STGB indices such as (031) or (012) but instead represent a complex GB structure composed of different GB planes as local segments.

We greatly appreciate the detailed observations of the reviewer, which have helped us improve the clarity and accuracy of both the text and figure presentation.

Changes made in response to the comment:

We have updated Fig. 3 to address the suggestion from the reviewer.

Page 7, modified Fig. 3:

Fig. 3. Characterization of secondary GB dislocations and their linkage to segregation patterns in the Fe-1 at.% W specimen... Cubic symbols in indicate the orientation of the grains and the red arrow indicates the misorientation rotation axis...

Page 7, line 175-178, added: “The character of GB1 corresponds to a geometrically curved boundary comprising a range of local orientations rather than an ideal $\Sigma 5$ symmetric tilt or twist GB. The color variation along GB1 reflects this orientation spread.”

Reviewer #2 Comment 3: The description of Figs. 3f, g say that the black arrows, associated with the STGBs, show significant deviations from the Skew normal distribution. This is not at all obvious to me. As a fractional difference between the data and the curve, I see places that deviate more. How do we know these deviations are not simply experimental uncertainty?

Response: We thank the reviewer for this valuable comment. Due to inherent limitations related to the statistical sampling area (which are practically unavoidable), the deviations (marked as "shoulders" by arrows) in Figs. 3f and g (Figs. 4f and g in the revised manuscript) may appear somewhat subtle. We have added further analysis and provided supporting evidence for the observed deviation. For example, as shown in Figs. 5a and 5b, the positions of the troughs and peaks for GB1 are located around 16 and 20 kJ/mol, and for GB2 at approximately 14 and 23 kJ/mol. These positions correspond precisely to the prominent shoulders marked by black arrows in Figs. 4f and 4g, offering additional evidence for the deviation of the segregation energy spectra from a skew-normal distribution.

Changes made in response to the comment:

Page 10-11, line 252-257, added: “It is important to note that the positions of troughs and peaks in Figs. 5a & b. For GB1, these are located around 16 and 20 kJ/mol, and for GB2 at 14 and 23 kJ/mol. These positions correspond precisely to the prominent shoulders marked by black arrows in Figs. 4f & g, providing side evidence for explaining the deviation of segregation energy spectra from the skew-normal distribution.”

Reviewer #2 Comment 4: p. 2. "planar defects" I recommend two-dimensional defects.

Response: We thank the reviewer for this suggestion. We have changed " planar defects" to "two-dimensional (2D) defects."

Changes made in response to the comment:

Page 1, line 19, modified: "...although defined as two-dimensional defects..."

Page 2, line 33-34, modified: "These grains meet at junctions to form piecewise two-dimensional defects known as grain boundaries (GBs)."

Reviewer #2 Comment 5: p. 2. "More specific in" I recommend more specifically in ...

Response: We thank the reviewer for this suggestion. We have changed "More specific in" to "more specifically in"

Changes made in response to the comment:

Page 2, line 48, modified: "More specifically in the model system FeW studied here..."

Reviewer #2 Comment 6: p. 2 "Real GB structures invariably break down into facets and regular patterns formed by secondary GB dislocations, to accommodate topological constraints [1, 2]. These curved surfaces" Are they facets or are they curved? They can't be both singular and rough.

Response: We thank the reviewer for requesting clarification on this important point, as it could indeed cause confusion. The curved surfaces are composed of planar segments with differing inclination. These facets are typically associated with the structure at the nanoscale. At the microscale, GBs appear curved with locally inclined GB planes; however, at the nanoscale, such inclined GB planes can be consisted of facets. Accordingly, we have revised the sentence based on the comment of the reviewer.

Changes made in response to the comment:

Page 2, line 65-68, added: "Real GB structures invariably break down into planar segments of differing inclination and regular patterns formed by secondary GB dislocations to accommodate topological constraints [1, 2]."

Reviewer #2 Comment 7: Fig. 1 and 2. How is the APT tip aligned with the film growth direction?

Response: We demonstrate how the APT tip aligns with the film growth direction in Supplementary Fig. 1. The coordination details are also presented in Figs. 2 and 3 (Figs. 1 and 2 in the original manuscript). Please refer to the captions of Figs. 2 and 3, where we have included the statement regarding the thin film growth direction in revised manuscript.

Changes made in response to the comment:

Page 6, Fig. 2 caption, added: "The X-axis represents the thin film growth direction in the same coordinate system in Supplementary Fig. 1."

Fig. 2. Correlative tomography characterization of the Fe-1 at.% W specimen ... The X-axis represents the thin film growth direction in the same coordinate system in Supplementary Fig. 1...

Page 7, Fig. 3 caption, added: "The X-axis represents the thin film growth direction in the same coordinate system in Supplementary Fig. 1." (Refer to our response to Reviewer #2, Comment 2, for the updated Fig. 3.)

Reviewer #3: This manuscript has conducted an excellent job on quantifying the relationship between secondary GB dislocations and their segregation energy spectra within a model Fe-W alloy. The authors claim that secondary GB dislocations can have an additional and, in some cases, even a much stronger effect on GB segregation than defect-free GBs. This is an interesting topic and the experimental data can well support the main conclusion of this paper. For the benefit of the readers, this manuscript needs a minor revision before acceptance for publication.

Response: We thank the reviewer for describing our work as an “excellent job” and the subject as “an interesting topic,” and for noting that the experimental data “well support” our conclusions. We appreciate these positive remarks and will incorporate the reviewer’s specific suggestions in the revised manuscript.

They are given below:

Reviewer #3 Comment 1: In Figures 3 B and C, the authors have quantitatively shown the interfacial solute excess of W segregation and the corresponding segregation energy across different grain boundaries. Nevertheless, the focus of this work is to highlight the effect of secondary GB dislocations on interfacial segregation, as driven by the GB curvatures. Thus, it is recommended that the grain boundary curvature, secondary dislocation density map, interfacial solute excess of W and the calculated segregation energy from the same region should be present together.

Response: We sincerely thank the reviewer for this valuable suggestion. As the reviewer correctly noted, our principal aim is to correlate secondary GB dislocations with the chemically heterogeneous segregation observed at GBs.

First, we refer the reviewer to our response to Reviewer #2 Comment 1, in which we explain that GB curvature manifests as local changes in GB-plane orientation. These variations are visualized in the orientation map of the GB-plane normal (see Fig. 3b and f). The virtual DF image of the same region further reveals a portion of the secondary GB dislocations (see Fig. 3c and g). Although the virtual DF image exposes some secondary GB dislocations, it cannot capture every dislocation; therefore, one cannot obtain the secondary GB-dislocation density directly from either the GB-plane-normal map or the virtual DF image.

To address this limitation, we have conducted additional theoretical analysis. We have added Fig. 5f, which correlates the segregation-energy line profile with GB inclination. We further analyzed the influence of GB-plane inclination and deviations in misorientation on the density of secondary GB dislocations using the Frank–Bilby framework. This theoretical analysis, combined with crystallographic data obtained from 4D-STEM tomography, provides an informed estimate of the configuration of secondary GB dislocations (see Fig. 5h). Figure 5i presents the corresponding segregation-energy modulation profile for this dislocation configuration.

We believe that the additional data in Figs. 5f–i effectively demonstrate the spatial correlation between GB curvature (described by local inclination angle ϕ), the misorientation deviation θ , the secondary dislocation density map, the interfacial solute excess of W (represented by the modulation factor in Fig. 5e), and the calculated segregation energy distribution. These results collectively highlight the relationship between the secondary GB dislocation distribution and the heterogeneous solute segregation behavior.

We direct the reviewer to our response to Reviewer #1 Comment 4a, where the corresponding modifications are also presented.

Reviewer #3 Comment 2: There remains a significant difference between the experimental GB image and the theoretical model. If possible, the AC-STEM is suggested to characterize the real GB structure at the atomic scale, as this would provide a more intuitive understanding of the atomic arrangement of secondary grain boundary dislocations.

Response: We appreciate the insightful suggestions from the reviewer. In response, we have added new results that characterize the GB structure at the atomic scale. Specifically, we now include atomic-resolution imaging data that directly reveal the presence and structure of secondary GB dislocations. These results are presented in the revised Figs. 1k and l and supplementary Fig. 5, which provide a direct understanding of the atomic arrangement at the GB. We believe this addition significantly strengthens the experimental foundation of our work and better supports our interpretations.

Changes made in response to the comment:

Page 5, line 130-137, added: “Figure 1k provides atomic-resolution images of the selected region in Fig. 1j, capturing the termination point of a secondary GB dislocation in the $\Sigma 5$ GB. The atomic columns around the GB are highlighted with red markers. We overlay the observed structure with a displacement shift complete (DSC) lattice and construct a Frank circuit, as shown in Fig. 1l (also see Supplementary Fig. 6 for the reference circuit), to determine the projected Burgers vector along the beam direction associated with the secondary GB dislocation.”

We also direct the reviewer to our response to Reviewer #1 Comment 1, for additional modifications related to this question.

Reviewer #3 Comment 3: I am interested about the spatial resolution error associated with the orientation maps of the local normal to the GB plane. Specifically, I would like to know whether such high spatial resolution can be achieved and, if so, how it is accomplished. Please provide a detailed explanation in the revised text.

Response: We appreciate the suggestion from the reviewer to clarify the explanation of spatial resolution errors in the orientation maps of the local normal to GB planes. In this study, we employed virtual DF slices along with alignment algorithms detailed in the Methods section to

reconstruct and analyze GBs in polycrystalline Fe–1 at.% W, focusing on determining the local normal to each GB plane.

As described in the Methods section, we first imported and processed the orientation and volume data to identify individual grains and their GBs. A marching cubes algorithm was then used to generate mesh representations of these grains and their interfaces. The normals of the mesh facets were calculated to determine the GB orientations, and by correlating these results with crystallographic orientation data, the mesh facets were color-coded according to their orientations. Because this 3D reconstruction method incorporates both volumetric information and in-plane details, it captures the approximate orientation of the GB plane. For example, as shown for GB3 in Supplementary Fig. 11, the reconstruction reveals local features along the GB that are consistent with the W segregation features observed in APT results.

The theoretical resolution of the 3D imaging is determined by the number of tilt images and the APT needle size, following the Crowther criterion [Measurement Science and Technology 29 (2018) 034005]. Based on the Crowther criterion, we estimate a theoretical spatial resolution of approximately 10 nm, which closely matches the spatial resolution obtained by comparing the 4DSTEM tomography reconstructed GB with that of the APT measurements.

Nonetheless, spatial resolution errors remain, as explained in the revised text in the Methods section. Projection misalignment, calibration errors, or both may occur, introducing spatial deviations during reconstruction via the Simultaneous Iterative Reconstruction Technique (SIRT). Because SIRT relies on an idealized forward model that assumes straight-line electron propagation, it neglects electron scattering and diffraction interactions, potentially leading to artifacts in the contrast of virtual DF images for individual grains. Additionally, noise in the projection data (arising from 4D-STEM detector acquisition) can be introduced and amplified during iterative processing. Insufficient angular coverage may further lead to missing information, causing blurring artifacts in the reconstruction.

Furthermore, because grain surface meshing involves Delaunay triangulation and Gaussian smoothing, interpolation is inevitably introduced into the mesh, resulting in a denser mesh than the theoretical spatial resolution predicted by the Crowther criterion. While this interpolation improves surface continuity, it also introduces additional uncertainty and reduces spatial precision at fine scales. Therefore, the resulting artificially high resolution should be interpreted with caution. We have clarified these limitations in the revised Methods section. Despite these limitations, the method still preserves sufficient geometric detail to identify the local GB normal, albeit at limited spatial resolution. Thus, in our study, instead of matching the precise orientation details of each mesh element to the corresponding locations of GB planes, we focus on capturing the microscopic curvature and orientation of the GB surface.

Changes made in response to the comment:

Page 16, line 443-452, added: “The spatial resolution of the 3D imaging is constrained by the number of tilt images and feature size, following the Crowther criterion [71], which estimates a 3D spatial resolution of approximately 10 nm. Accordingly, in this study, features smaller than 10 nm could not be resolved. Given that spatial resolution errors may be introduced during

4DSTEM data collection, alignment, and reconstruction, as well as through meshing and the inevitable smoothing algorithms, we focus on capturing the microscopic curvature and orientation of the GB surface as a reference, rather than precisely matching the orientation of every mesh element to the corresponding GB planes.”